# MULTI-VIEW REPRESENTATION IS WHAT YOU NEED FOR POINT-CLOUD PRE-TRAINING

**Siming Yan[†], Chen Song[†], Youkang Kong[‡], Qixing Huang[†]**

[†]The University of Texas at Austin, [‡]Microsoft Research Asia
`{siming, song, huangqx}@cs.utexas.edu, {kykdqs}@gmail.com`

## ABSTRACT

A promising direction for pre-training 3D point clouds is to leverage the massive amount of data in 2D, whereas the domain gap between 2D and 3D creates a fundamental challenge. This paper proposes a novel approach to point-cloud pre-training that learns 3D representations by leveraging pre-trained 2D networks. Different from the popular practice of predicting 2D features first and then obtaining 3D features through dimensionality lifting, our approach directly uses a 3D network for feature extraction. We train the 3D feature extraction network with the help of the novel 2D knowledge transfer loss, which enforces the 2D projections of the 3D feature to be consistent with the output of pre-trained 2D networks. To prevent the feature from discarding 3D signals, we introduce the multi-view consistency loss that additionally encourages the projected 2D feature representations to capture pixel-wise correspondences across different views. Such correspondences induce 3D geometry and effectively retain 3D features in the projected 2D features. Experimental results demonstrate that our pre-trained model can be successfully transferred to various downstream tasks, including 3D shape classification, part segmentation, 3D object detection, and semantic segmentation, achieving state-of-the-art performance.

## 1 INTRODUCTION

The rapid development of commercial data acquisition devices (Daneshmand et al., 2018) and point-based deep learning networks (Qi et al., 2017a;b; Choy et al., 2019) has led to a growing research interest in models that can directly process 3D point clouds without voxelization. Remarkable success has been achieved in various applications, including but not limited to object detection (Misra et al., 2021; Liu et al., 2021b; Wang et al., 2022a), segmentation (Qian et al., 2022; Tang et al., 2022; Zhao et al., 2021; Yan et al., 2021), and tracking (Qi et al., 2020; Zheng et al., 2021; Shan et al., 2021).

Despite the significant advances in 3D point cloud processing, acquiring task-specific 3D annotations is a highly expensive and severely limited process due to the geometric complexity. The shortage of data annotations highlights the need for adapting pre-training paradigms. Instead of training the deep network from randomly initialized weights, prior work suggests that pre-training the network on a relevant but different pre-task and later fine-tuning the weights using task-specific labels often leads to superior performance. In natural language processing (Devlin et al., 2018) and 2D vision (He et al., 2022; Radford et al., 2015; Doersch et al., 2015; He et al., 2020; Zhuang et al., 2021; 2019), pre-trained models are the backbones of many exciting applications, such as real-time chatbots (Touvron et al., 2023; OpenAI, 2023) and graphic designers (Meng et al., 2021; Wang et al., 2022b). However, pre-training on point clouds has yet to demonstrate a universal performance improvement. From-scratch training remains a common practice in 3D vision.

Initial attempts towards 3D point-cloud pre-training primarily leverage contrastive learning (Chopra et al., 2005), especially when the point clouds are collected from indoor scenes (Xie et al., 2020; Rao et al., 2021; Liu et al., 2021a; Zhang et al., 2021; Chen et al., 2022). However, the broad application of contrastive learning-based pre-training techniques is impeded by the requirement of large batch sizes and the necessity to carefully define positive and negative pairs. In contrast to natural language processing and 2D vision, pre-training on 3D point clouds presents two unique challenges.

First, the data is extremely scarce, even without annotations. Public 3D datasets are orders of magnitude smaller than 2D image datasets. Second, the lack of data annotations necessitates 3D pre-training methods to adhere to the self-supervised learning paradigm. Without strong supervision, pre-task design becomes particularly crucial in effective knowledge acquisition.

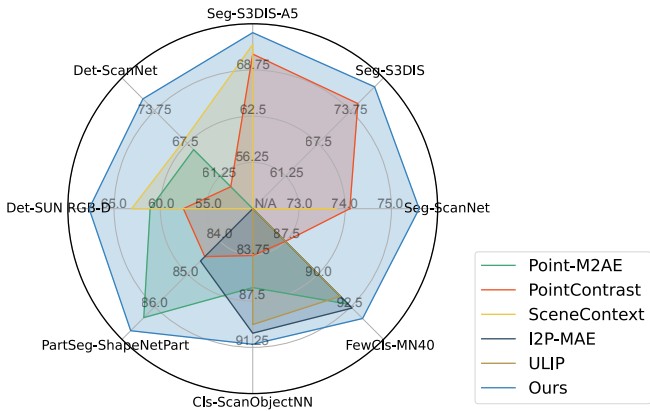

Figure 1: Our model (blue) achieves state-of-the-art performance across a broad range of tasks at both the scene and shape levels. The distance to the origin indicates the task result.

Figure 2 illustrates our novel approach to 3D pre-training, which is designed to address the aforementioned challenges. Our key idea is to leverage pre-trained 2D networks in the image domain to enhance the performance of 3D representation learning while bridging the 2D-3D domain gap. We begin by applying a 3D-based network to an input point cloud with $M$ points, which generates a 3D feature volume represented by an $M \times C$-dimensional embedding, where $C$ is the length of the per-point feature vectors. To compensate for data scarcity, we align the 3D feature volume with features predicted by a 2D encoder trained on hundreds of millions of images. To this end, we project the 3D feature volume into pixel embeddings and obtain $H \times W \times C$ image-like feature maps in 2D. We then use pre-trained image encoders to extract hierarchical features from corresponding RGB images and train a similarly structured encoder on projected 2D maps to enforce feature alignment. To ensure geometrical meaningfulness, we project the same 3D feature volume into multiple camera views and perform hierarchical feature alignment in each individual view. Our pipeline is generic enough to support all major 2D pre-trained models (e.g., CLIP (Radford et al., 2021), SAM (Kirillov et al., 2023), DINOv2 (Oquab et al., 2023)). In our experiments, DINOv2 exhibits the best performance.

One issue with the approach described above is that the 3D feature learning may overfit to the pre-trained 2D networks. Consequently, it could potentially discard 3D features that are critical for 3D recognition but not well-captured during image-based 2D pre-training. To address this issue, we introduce an auxiliary pre-task to predict 2D multi-view pixel-wise correspondences from pairs of projected 2D feature maps. As such correspondences induce 3D depth information, the learned feature representations are forced to capture 3D signals.

Our approach is fundamentally different from existing work based on multi-view representations (Su et al., 2015; Kalogerakis et al., 2017; Kanezaki et al., 2018; Wei et al., 2020; Kundu et al., 2020; Hamdi et al., 2021; 2023), which either aggregate image-based analysis results or lift 2D feature maps into 3D for understanding. Our approach employs a 3D network for feature extraction trained with the additional help of **pre-trained** 2D models. While multi-view based 3D understanding approaches (Su et al., 2015; Kalogerakis et al., 2017) tremendously benefit from the rich experience in 2D vision and have long dominated the design of shape classification networks when 3D data is limited, recent 3D-based understanding networks (Riegler et al., 2017; Qi et al., 2017a;b; Wang et al., 2017) are shown to outperform their multi-view counterparts. The superior performance is the result of enhanced architecture design and the clear advantage of directly learning 3D patterns for geometry understanding. In our setting, the 3D feature extraction network provides implicit regularization on the type of features extracted from pre-trained 2D models. We seek to combine the advantage of multi-view representations and the geometric capacity of 3D networks, while leveraging the rich knowledge embedded in pre-trained 2D networks.

In summary, we make the following contributions:

- We formulate point-cloud pre-training as learning a multi-view consistent 3D feature volume.

- To compensate for data scarcity, we leverage pre-trained 2D image-based models to supervise 3D pre-training through perspective projection and hierarchical feature-based 2D knowledge transfer.

- To prevent overfitting to pre-trained 2D networks, we develop an auxiliary pre-task where the goal is to predict the multi-view pixel-wise correspondences from the 2D pixel embeddings.

- We conduct extensive experiments to demonstrate the effectiveness of our approach across a wide range of downstream tasks, including shape classification, part segmentation, 3D detection, and semantic segmentation, achieving consistent improvement over baselines (Figure 1).

## 2 RELATED WORK

**Point-cloud pre-training**    The success of point-based deep neural networks demonstrates the potential for machine learning models to directly perceive point clouds. Point-based pre-training refers to the practice of *pre-training* the point-based prediction network on one or more *pre-tasks* before fine-tuning the weights on *downstream tasks*. The expectation is that the knowledge about pre-tasks will be transferred to downstream tasks, and the network can achieve better performance than random parameter initialization. We refer readers to (Xiao et al., 2023) for a comprehensive survey of the field, in which we highlight one important observation, stating that point-based pre-training "still lags far behind as compared with its counterparts", and training from-scratch "is still the prevalent approach". The challenges and opportunities call for innovations in point-based pre-training methods.

*Shape-level pre-training.* Self-reconstruction is a popular pre-task at the shape level, where the network encodes the given point cloud as representation vectors capable of being decoded back to the original input data (Yang et al., 2018; Sauder & Sievers, 2019). To increase the pre-task difficulty, it is common for self-reconstruction-based methods to randomly remove a certain percentage of the points from the input (Wang et al., 2021; Pang et al., 2022; Zhang et al., 2022a; Yu et al., 2022; Yan et al., 2023b;a). In addition to self-reconstruction, (Rao et al., 2020) develop a multi-task approach that unifies contrastive learning, normal estimation, and self-reconstruction into the same pipeline. There are also attempts to connect 3D pre-training to 2D (Afham et al., 2022; Xu et al., 2021; Dong et al., 2022; Zhang et al., 2022b), or to explore the inverse relationship (Hou et al., 2023). For example, (Dong et al., 2022) utilize pre-trained 2D transformers as cross-modal teachers. (Zhang et al., 2022b) leverage self-supervised pre-training and masked autoencoding to obtain high-quality 3D features from 2D pre-trained models. However, these methods focus on synthetic objects, leading to significant performance degradation in downstream tasks using real data from indoor scenes.

*Scene-level pre-training.* Existing scene-level pre-training methods focus on exploiting the contrastive learning paradigm. As the first pre-training method with demonstrated success at the scene level, PointContrast (Xie et al., 2020) defines the contrastive loss using pairs of 3D points across multiple views. RandomRoom (Rao et al., 2021) exploits synthetic datasets and defines the contrastive loss using pairs of CAD objects instead. Experimentally, RandomRoom additionally confirms that synthetic shape-level pre-training is beneficial to real-world scene-level downstream tasks. DepthContrast (Zhang et al., 2021) simplifies PointContrast by proposing an effective pre-training method requiring only single-view depth scans. 4DContrast (Chen et al., 2022) further includes temporal dynamics in the contrastive formulation, where the pre-training data comprises sequences of synthetic scenes with objects in different locations. In contrast to these approaches, we focus on utilizing pre-trained 2D networks to boost the performance of 3D pre-training without leveraging contrastive learning.

## 3 METHOD

We present our pre-training approach pipeline, denoted as **MVNet** in the following sections. As illustrated in Figure 2, given a pair of RGB-D scans, we first project the 2D pixels into 3D point clouds using camera parameters. We then extract the point-cloud feature volumes using the feature encoding network (Section 3.1). Subsequently, we project the feature volume onto two different views to generate 2D feature maps (Section 3.2). We design the 2D knowledge transfer module to learn from large-scale 2D pre-trained models (Section 3.3). Finally, we utilize the multi-view consistency module to ensure the agreement of different view features (Section 3.4), promoting that the 3D feature encoding network extracts 3D-aware features from pre-trained 2D image models. Finally, the weights of the feature encoding network are transferred to downstream tasks for fine-tuning.

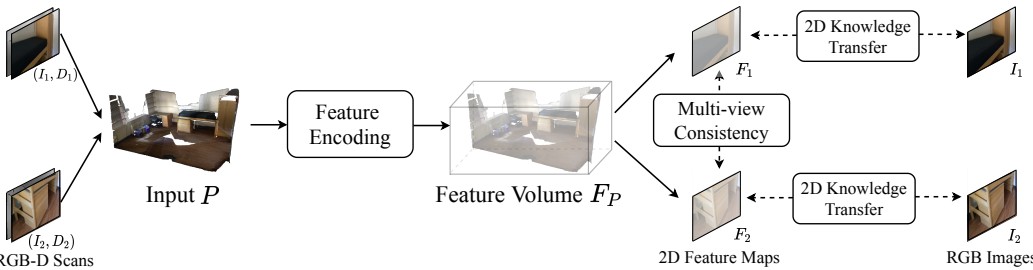

Figure 2: **Approach overview.** We used complete and semi-transparent point clouds to represent the input $P$ and the feature volume $F_P$ for better visualization. We encourage readers to frequently reference to this figure while reading Section 3.

### 3.1 FEATURE ENCODING NETWORK

Let $(I_1, D_1)$ and $(I_2, D_2)$ be two RGB-D scans from the same scene, where $I_1$ and $I_2$ denote the RGB images, and $D_1$ and $D_2$ represent the depth maps. Using camera parameters, we project the RGB-D scans into a colored point cloud $P \in \mathbb{R}^{M \times 6}$ with $M$ points, with the first three channels representing coordinates and the remaining three channels representing RGB values. The feature encoding network of MVNet takes $P$ as input and outputs a dense feature field $F_P \in \mathbb{R}^{M \times C}$, where each point is associated with a feature vector of dimension $C$.

The network architecture adopts the design of Sparse Residual U-Net (SR-UNet) (Choy et al., 2019). Our network includes 21 convolution layers for the encoder and 13 convolution layers for the decoder, where the encoder and the decoder are connected with extensive skip connections. Specifically, the input point cloud $P$ is first voxelized into $M'$ voxels, yielding a grid-based representation $V \in \mathbb{R}^{M' \times 6}$. In our implementation, we use the standard Cartesian voxelization with the grid length set to 0.05 m. The output of the network is a set of $C$-dimensional per-voxel features, $F_V \in \mathbb{R}^{M' \times C}$, jointly generated by the encoder and the decoder. Due to space constraints, we refer interested readers to the supplementary material for implementation details.

Next, we interpolate the per-voxel features $F_V$ to obtain a $C$-dimensional feature vector for each of $M$ points in the input point cloud. This is achieved via the classical KNN algorithm based on point-to-point distances. As shown in the middle of Figure 2, the resulting $F_P$ is our point-cloud feature volume, awaiting further analysis and processing.

Importantly, unlike existing work that utilizes multi-view representations from 2D feature extraction networks, MVNet exploits a 3D feature extraction network. As a result, MVNet has the intrinsic ability to capture valuable 3D patterns for 3D understanding.

### 3.2 POINT-CLOUD FEATURE VOLUME PROJECTION

The key idea of MVNet is to learn 3D feature representations through 2D projections, enabling us to leverage large-scale pre-trained models in 2D. To this end, we proceed to project the feature volume $F_P$ back onto the two input views, generating 2D feature maps, $F_1$ and $F_2$, both with dimensions $H \times W \times C$. Here, $H \times W$ is the spatial resolution of the input.

The projection operation uses the one-to-one mapping between each pixel in each input view and the corresponding 3D point in the concatenated 3D point cloud $P$. We refer readers to the supplementary material for the implementation details and the conversion formula between 2D and 3D coordinates.

### 3.3 2D KNOWLEDGE TRANSFER MODULE

The first feature learning module of MVNet, $\mathcal{F}_{2d}$, aims to transfer knowledge from large-scale 2D pre-trained models. To this end, we consider each input RGB image $I_i$ and the corresponding projected 2D feature map $F_i$. Our goal is to train $F_i$ using a pre-trained 2D model $f_p$ that takes $I_i$ as input. Our implementation uses ViT-B (Dosovitskiy et al., 2020) as the network architecture. More details are provided in Section 4.3.

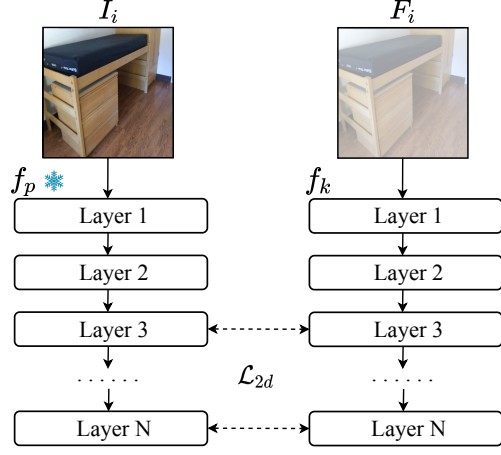

The technical challenge here lies in the semantic differences between $F_i$ and $I_i$. As illustrated in Figure 3, we introduce an additional network $f_k$ that takes $F_i$ as input. $f_k$ shares the same network architecture as $f_p$ except the first layer, which is expanded channel-wise to accommodate the $C$-dimensional input. Let $N$ be the number of layers in $f_k$ and $f_p$. We define the training loss as

$$\mathcal{L}_{2d} = \sum_{j=3}^{N} \|B_j^{\text{pre-trained}} - B_j^{\text{2D}}\|_2^2 \qquad (1)$$

where $B_j^{\text{pre-trained}}$ denotes the output feature map of the $j^{\text{th}}$ block in the pre-trained 2D model $f_p$, and $B_j^{\text{2D}}$ denotes the output feature map of the $j^{\text{th}}$ block in $f_k$. The objective of the 2D knowledge transfer process is to minimize $\mathcal{L}_{2d}$, thereby enabling $f_k$ to learn the hierarchical features through knowledge transfer from the pre-trained 2D model. We add the loss starting from the third layer, which experimentally leads to the best performance.

Figure 3: **2D knowledge transfer module.** $f_p$ is completely frozen during training. For simplicity, we represent the $C$-dimensional feature map $F_i$ using a semi-transparent image.

During training, the weights of the 2D pre-trained model $f_p$ are unchanged, whereas $f_k$ and the feature encoding network are jointly optimized. In our implementation, we select ViT-B (Dosovitskiy et al., 2020) as the network backbone for both 2D neural network $f_k$ and $f_p$. We take the pre-trained weights of DINOv2 (Oquab et al., 2023) for $f_p$. DINOv2 has been demonstrated to generate features with strong transferability. We refer readers to Section 4.3 for a detailed analysis of these design choices through extensive ablation studies.

### 3.4 MULTI-VIEW CONSISTENCY MODULE

The goal of the previous module is to leverage pre-trained 2D networks. However, pre-trained 2D networks only contain feature representations that are primarily suitable for 2D tasks. A complete reliance on the 2D knowledge transfer module encourages the 3D feature encoding network to discard geometric cues that are important for 3D recognition. To address this issue, we introduce a novel multi-view consistency module $\mathcal{F}_m$, where the goal is to use the projected features, $F_1$ and $F_2$, to predict dense correspondences between the two corresponding input images. Prior research has demonstrated that dense pixel-wise correspondences between calibrated images allows the faithful recovery of 3D geometry (Zhou et al., 2016). Therefore, enforcing ground-truth correspondences to be recoverable from $F_1$ and $F_2$ prevents the learned feature volume from discarding 3D features.

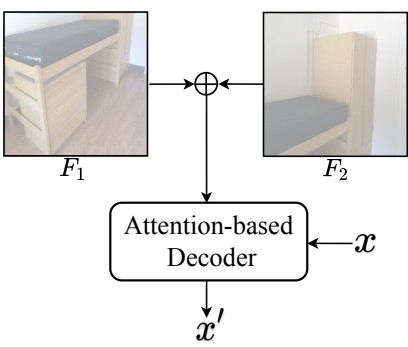

Figure 4: **Illustration of multi-view consistency module.** $\oplus$ denotes side-by-side concatenation.

Our dense correspondence module adopts a cross attention-based transformer decoder $\mathcal{D}$. As depicted in Figure 4, we concatenate the two views' feature maps side by side, forming a feature map $F_c$. The context feature map $F_c$ is fed into a cross attention-based transformer decoder $\mathcal{D}$, along with the query point $x$ from the first view $I_1$. Finally, we process the output of the transformer decoder with a fully connected layer to obtain our estimate for the corresponding point, $x'$, in the second view $I_2$:

$$x' = \mathcal{F}_m(x|F_1, F_2) = \mathcal{D}(x, F_1 \oplus F_2) \qquad (2)$$

Following (Jiang et al., 2021), we design the loss for correspondence error and cycle consistency:

$$\mathcal{L}_m = \|x^{gt} - x'\|_2^2 + \|x - \mathcal{F}_m(x'|F_1, F_2)\|_2^2 \qquad (3)$$

where $x^{gt}$ denotes the ground truth corresponding point of $x$ in the second view $I_2$.

| Method | Backbone | S3DIS Area 5 | | | S3DIS 6-Fold | | | ScanNet |
|--------|----------|------|------|------|------|------|------|------|
| | | OA | mAcc | mIoU | OA | mAcc | mIoU | mIoU |
| Jigsaw (Sauder & Sievers, 2019) | DGCNN | 82.8 | - | 52.1 | 84.4 | - | 56.6 | - |
| OcCo (Wang et al., 2020) | DGCNN | 85.9 | - | 55.4 | 85.1 | - | 58.5 | - |
| STRL (Huang et al., 2021) | DGCNN | 82.6 | - | 51.8 | 84.2 | - | 57.1 | - |
| MM-3DScene (Xu et al., 2023) | PointTrans | - | 78.0 | 71.9 | - | - | - | 72.8 |
| PointContrast (Xie et al., 2020) | SR-UNet | - | 77.0 | 70.9 | - | - | - | 74.1 |
| DepthContrast (Zhang et al., 2021) | SR-UNet | - | - | 70.6 | - | - | - | 71.2 |
| SceneContext (Hou et al., 2021) | SR-UNet | - | - | 72.2 | - | - | - | 73.8 |
| MVNet | SR-UNet | **91.7**(91.6) | **79.5**(79.3) | **73.8**(73.3) | **91.8**(91.7) | **86.3**(86.2) | **78.3**(78.1) | **75.6**(75.2) |

Table 1: **Comparison of semantic segmentation results with pre-training methods.** We show a comprehensive evaluation across all the benchmarks. The average result of 3 runs is given in parentheses.

Combing (1) and (3), the overall loss function is defined as:

$$\mathcal{L} = \mathcal{L}_{2d} + \lambda \mathcal{L}_m \qquad (4)$$

where $\lambda = 0.5$ is the trade-off parameter.

## 4 EXPERIMENTAL ANALYSIS

### 4.1 PRE-TRAINING SETTING

**Data preparation** We choose ScanNet (Dai et al., 2017) as the pre-training dataset, which contains approximately 2.5M RGB-D scans from 1,513 indoor scenes. Following (Qi et al., 2019), we downsample 190K RGB-D scans from 1,200 video sequences in the training set. For each scan, we build the scan pair by taking it as the *first* scan $(I_1, D_1)$ and select five other frames with an overlap ratio between $[0.4, 0.8]$ as the *second* scan $(I_2, D_2)$ candidates. We also evaluate our model at the shape level. We pre-trained our model on Objaverse (Deitke et al., 2023), which contains approximately 800K real-world 3D objects. We also pre-trained our model on ShapeNet (Chang et al., 2015) for fair comparison with previous approaches. For each 3D object, we use Blender (Kent, 2015) to render 6 images with their camera angles spaced equally by 60 degrees. At each training step, we randomly choose two consecutive images for pre-training.

**Network training** Our pre-training model is implemented using PyTorch, employing the AdamW optimizer (Loshchilov & Hutter, 2017) with a weight decay of $10^{-4}$. The learning rate is set to $10^{-3}$. The model is trained for 200 epochs on eight 32GB Nvidia V100 GPUs, with 64 as the batch size.

### 4.2 DOWNSTREAM TASKS

The goal of pre-training is to learn features that can be transferred effectively to various downstream tasks. In the following experiments, we adopt the **supervised fine-tuning strategy**, a popular way to evaluate the transferability of pre-trained features. Specifically, we initialize the network with pre-trained weights as described in Section 3 and fine-tune the weights for each downstream task.

**Semantic segmentation** We use the original training and validation splits of ScanNet (Dai et al., 2017) and report the mean Intersection over Union (IoU) on the validation split. We also evaluate on S3DIS (Armeni et al., 2017) that has six large areas, where one area is chosen as the validation set, and the remaining areas are utilized for training. We report the Overall Accuracy (OA), mean Accuracy (mAcc), and mean IoU for Area-5 and the 6-fold cross-validation results.

We adopt SR-UNet (Choy et al., 2019) as the semantic segmentation architecture and add a head layer at the end of the feature encoding network for semantic class prediction. Although it is intuitive to transfer both the encoder and decoder pre-trained weights in segmentation fine-tuning, we observe that only transferring the encoder pre-trained weights results in a better performance. We fine-tune our model using the SGD+momentum optimizer with a batch size of 48, 10,000 iterations, an initial learning rate of 0.01, and a polynomial-based learning rate scheduler with a power factor of 0.9. The same data augmentation techniques in (Chen et al., 2022) are used.

As demonstrated in Table 1, on both datasets, we achieve the best performance compared to other pre-training methods, with +1.5 val mIoU improvement on ScanNet compared to the second-best method. On the S3DIS dataset, we report +1.6 mIoU improvement on the Area 5 validation set.

| Method | Detection Architecture | ScanNet | | SUN RGB-D | |
|---|---|---|---|---|---|
| | | mAP@0.25 | mAP@0.5 | mAP@0.25 | mAP@0.5 |
| STRL (Huang et al., 2021) | VoteNet | 59.5 | 38.4 | 58.2 | 35.0 |
| Point-BERT (Yu et al., 2022) | 3DETR | 61.0 | 38.3 | - | - |
| RandomRooms (Rao et al., 2021) | VoteNet | 61.3 | 36.2 | 59.2 | 35.4 |
| MaskPoint (Liu et al., 2022) | 3DETR | 63.4 | 40.6 | - | - |
| PointContrast (Xie et al., 2020) | VoteNet | 59.2 | 38.0 | 57.5 | 34.8 |
| DepthContrast (Zhang et al., 2021) | VoteNet | 62.1 | 39.1 | 60.4 | 35.4 |
| MM-3DScene (Xu et al., 2023) | VoteNet | 63.1 | **41.5** | 60.6 | 37.3 |
| SceneContext (Hou et al., 2021) | VoteNet | - | 39.3 | - | 36.4 |
| 4DContrast (Chen et al., 2022) | VoteNet | - | 40.0 | - | 38.2 |
| MVNet | VoteNet | **64.0**(63.4) | **41.5**(41.0) | **62.0**(61.7) | **39.6**(39.2) |
| MVNet | CAGroup3D | **76.0**(75.7) | **62.4**(61.8) | **67.6**(67.2) | **51.7**(51.2) |

Table 2: **Comparison of 3D object detection results with pre-training methods.** We show mean Average Precision (mAP) across all semantic classes with 3D IoU thresholds of 0.25 and 0.5. The average result of 3 runs is given in parentheses.

**3D object detection** ScanNet (Dai et al., 2017) contains instance labels for 18 object categories, with 1,201 scans for training and 312 for validation. SUN RGB-D (Song et al., 2015) comprises of 5K RGB-D images annotated with amodal, 3D-oriented bounding boxes for objects from 37 categories.

We employ the encoder of the SR-UNet (Choy et al., 2019) as the detection encoder and transfer the pre-trained weights for network initialization. For the decoder design of 3D detection, we consider VoteNet (Qi et al., 2019) and CAGroup3D (Wang et al., 2022a). VoteNet is a classical 3D detection approach commonly used by existing pre-training methods (Xie et al., 2020; Zhang et al., 2021; Rao et al., 2021; Huang et al., 2021). However, VoteNet's performance lags behind the state-of-the-art. Therefore, we also evaluate our approach on CAGroup3D, which offers a stronger decoder architecture and delivers state-of-the-art 3D detection performance. During training, for the VoteNet setting, we fine-tune our model with the Adam optimizer (the batch size is 8, and the initial learning rate is 0.001). The learning rate is decreased by a factor of 10 after 80 and 120 epochs. For the CAGroup3D setting, we decrease the learning rate at 80 and 110 epochs.

As shown in Table 2, on ScanNet, our approach (MVNet) improves VoteNet by +4.5/+3.1 points on mAP@0.25 and mAP@0.5, outperforming other pre-training approaches. When replacing VoteNet with CAGroup3D, MVNet further elevates the mAP@0.25/mAP@0.5 to 76.0/62.4. On the SUN RGB-D dataset, MVNet also exhibits consistent improvements. Specifically, MVNet achieves +3.8/+4.6 points on mAP@0.25 and mAP@0.5 under the VoteNet backbone and 67.6/51.7 points on mAP@0.25 and mAP@0.5 under the CAGroup3D backbone.

**3D shape part segmentation** In addition to scene-level downstream tasks, we also evaluate MVNet at the shape level. Our experiments on 3D shape part segmentation utilizes the ShapeNetPart (Yi et al., 2016), which comprises 16,880 models across 16 shape categories. We adopt the standard evaluation protocol as described by (Qi et al., 2017b), sampling 2,048 points from each shape. We report the evaluation metrics in terms of per-class mean Intersection over Union (cls. mIoU) and instance-level mean Intersection over Union (ins. mIoU).

To make a fair comparison, in accordance with (Yu et al., 2022), we employ a standard transformer network with a depth of 12 layers, a feature dimension of 384, and 6 attention heads, denoted as MVNet-B. The experimental results are summarized in Table 3. With the same network backbone, our approach demonstrates consistent improvements. Additionally, more results involving pre-training approaches with different network backbones can be found in the supplementary material.

**3D shape classification** We also conduct experiments on the real-world ScanObjectNN dataset (Uy et al., 2019), which contains approximately 15,000 scanned objects distributed across 15 classes. The dataset is partitioned into three evaluation splits: OBJ-BG, OBJ-ONLY, and PB-T50-RS, where PB-T50-RS is considered the most challenging one. We employ the same transformer-based classification network (MVNet-B) as described by (Yu et al., 2022). The experimental outcomes, presented in Table 3, reveal consistent performance gains across all three test sets.

**Model scalability** The scalability of a model is a crucial factor when evaluating the effectiveness of a pre-training approach. Unfortunately, many previous methods overlook this aspect. The commonly used transformer architecture has limited capacity due to the small number of parameters. To further

| Method | Dataset | ScanObjectNN | | | ShapeNetPart | |
|---|---|---|---|---|---|---|
| | | OBJ-BG | OBJ-ONLY | PB-T50-RS | ins. mIoU | cls. mIoU |
| Transformer[†]  (Yu et al., 2022) | - | 79.9 | 80.6 | 77.2 | 85.1 | 83.4 |
| PointBERT (Yu et al., 2022) | ShapeNet | 87.4 | 88.1 | 83.1 | 85.6 | 84.1 |
| MaskDiscr (Liu et al., 2022) | ShapeNet | 89.7 | 89.3 | 84.3 | 86.0 | 84.4 |
| MaskSurfel (Zhang et al., 2022c) | ShapeNet | 91.2 | 89.2 | 85.7 | 86.1 | 84.4 |
| ULIP (Xue et al., 2023) | ShapeNet | 91.3 | 89.4 | 86.4 | - | - |
| PointMAE (Pang et al., 2022) | ShapeNet | 90.0 | 88.3 | 85.2 | 86.1 | - |
| MVNet-B | ShapeNet | **91.4** | **89.7** | **86.7** | **86.1** | **84.8** |
| MVNet-B | Objaverse | **91.5** | **90.1** | **87.8** | **86.2** | **84.8** |
| MVNet-L | Objaverse | **95.2** | **94.2** | **91.0** | **86.8** | **85.2** |

Table 3: **Performance comparison with other pre-training approaches on shape-level downstream tasks.** All the methods and MVNet-B use the same transformer backbone architecture. [†] represents the *from scratch* results and all other methods represent the *fine-tuning* results using pretrained weights.

substantiate the efficacy of our proposed approach, we improve the network's feature dimension to 768 and double the number of attention heads to 12. As evidenced by the results in Table 3, our scaled model, MVNet-L, attains significant improvements across all evaluated settings.

### 4.3 ABLATION STUDY

We proceed to present an ablation study to justify our design choices. Due to the space constraint, we focus on the semantic segmentation results using the ScanNet dataset.

**Choice of 2D pre-trained models** As one of the most popular backbone architectures, ViT (Dosovitskiy et al., 2020) has two variants: ViT-S and ViT-B. ViT-S (Small) contains 12 transformer blocks, each of which has 6 heads and 384 hidden dimensions. ViT-B (Base) contains 12 transformer blocks, each of which has 12 heads and 768 hidden dimensions. We also compare four different pre-training methods. As shown in Figure 5, 'ViT-X','CLIP-X', 'SAM-X', and 'DINOv2-X' represent pre-training the model by ImageNet-1K classification (Deng et al., 2009), the model with the CLIP approach (Radford et al., 2021), the model with the SAM approach (Kirillov et al., 2023), and the model with DINOv2 (Oquab et al., 2023), respectively. We observe that all the models show improvements compared with training from scratch (72.2 mIoU). Larger

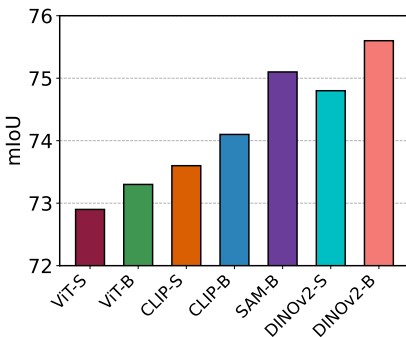

Figure 5: **Comparison of different 2D pre-trained models.**

models (X-B) show consistent improvements compared with smaller models (X-S). DINOv2-B yields the best performance in terms of semantic segmentation accuracy. This demonstrates the advantage of leveraging DINOv2's strong transferability properties in our method.

**Loss design** Table 4a demonstrates the performance of our method with different loss configurations. We investigate the effectiveness of each loss component: $\mathcal{L}_{2d}$ (2D knowledge transfer loss) and $\mathcal{L}_m$ (multi-view consistency loss). The table shows that using either loss independently leads to an improvement in performance. However, combining both losses yields the best results, validating the importance of incorporating both 2D knowledge transfer and multi-view consistency in our approach.

**Masking ratio** During pre-training, our approach involved masking the input point cloud guided by the corresponding RGB image. Following the conventional practice in masked autoencoder-based pre-training (He et al., 2022), we divided the image into regular non-overlapping patches. Then we sample a subset of patches and mask the remaining ones. The remaining part is projected onto the point cloud, forming the input for our method. We employ this masking operation as a data augmentation technique, which assists in the development of more robust features during pre-training. As evidenced in Table 4b, an appropriate masking operation (i.e., 30%) during pre-training could enhance performance in downstream tasks.

**View overlap ratio** In Table 4c, we examine the impact of the view overlap ratio on the performance of our method. The view overlap ratio refers to the percentage of a shared field of view between the

| (a) **Loss design** | | | (b) **Masking ratio** | | (c) **View overlap** | | (d) **View number** | | |
|---|---|---|---|---|---|---|---|---|---|
| $\mathcal{L}_{2d}$ | $\mathcal{L}_m$ | ScanNet | Ratio (%) | ScanNet | Overlap (%) | ScanNet | Num | $\mathcal{L}_{2d}$ | $\mathcal{L}_m$ |
| - | - | 72.2 | 10 | 74.8 | 20 | 74.7 | 1 | 74.5 | 72.9 |
| √ | - | 74.7 | 30 | **75.6** | 40 | 75.1 | 2 | 74.6 | 73.1 |
| - | √ | 73.6 | 60 | 74.2 | 60 | 75.3 | 4 | 74.6 | 73.3 |
| √ | √ | **75.6** | 80 | 73.5 | 80 | **75.4** | 5 | **74.7** | **73.6** |

Table 4: **Ablation studies of our design choices**. We report the best val mIoU result on ScanNet. Please refer to Section 4.3 for a detailed analysis.

two views used for multi-view consistency. It is important to note that in this experiment, we only use **one** candidate second view. Compared to the results without the multi-view consistency loss (74.7 mIoU), overlaps between 40% and 80% all result in improvements. However, when the overlap is too small, specifically, at 20%, we fail to observe any improvement. We believe this is because when the overlap is insufficient, multi-view correspondences become too sparse, obtaining very limited 3D signals. As a result, our default setting selects views with overlaps in the range of $[40\%, 80\%]$ as candidate second views, which ensures a balance between shared and unshared information for robust model performance.

**Number of views** In Table 4d, we investigate the impact of using different numbers of candidate second views. We experiment with 1 view, 2 views, 4 views, and 5 views. $\mathcal{L}_{2d}$ denotes only adding 2D knowledge transfer loss, while $\mathcal{L}_m$ represents only adding multi-view consistency loss. We find that incorporating more views leads to improved performance for both modules. However, it shows minor performance variations for 2D knowledge transfer module. Our interpretation is that since the loss is applied on each view independently, the different number of candidate views exert minimal influence. Note that we did not experiment with more views as we observed that when the number of views exceeds 5, some candidates have a very small overlap with the first scan, which, as proven in the previous paragraph, can impede the learning of the multi-view consistency module.

**2D knowledge transfer loss** We proceed to investigate the optimal locations to enforce the 2D knowledge transfer loss. We explore the impact of incorporating the loss starting from different layers of $f_k$. As shown in Figure 6, adding the feature-based loss starting from the third layer yields the best performance in terms of semantic segmentation val mIoU on ScanNet. When adding the loss on the first and second layers, the model's performance is negatively impacted. We believe this is because the inputs of the 2D pre-trained model and the 2D neural network differ. Forcefully adding the loss on the early layers may not be beneficial. Furthermore, the performance drops when the loss is added starting from the fourth layer and beyond. This demonstrates that only regularizing the last few layers weakens the knowledge transferred from the pre-trained networks.

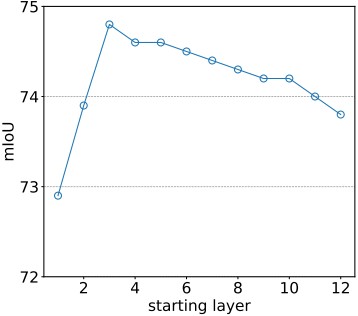

Figure 6: **Comparison of 2D knowledge transfer loss on different starting layers.**

## 5 CONCLUSION AND LIMITATION

In this paper, we present a novel approach to 3D point-cloud pre-training. We leverage pre-trained 2D networks to supervise 3D pre-training through perspective projection and hierarchical feature-based 2D knowledge transfer. We also introduce a novel pre-task to enhance the geometric structure through correspondence prediction. Extensive experiments demonstrate the effectiveness of our approach.

Potential avenues for future work include exploring additional pre-training data, such as NYUv2 (Couprie et al., 2013), and investigating the effect of increasing the size of the feature encoding network. Furthermore, we plan to extend our approach to outdoor settings. MVNet opens up new avenues for 3D point-cloud pre-training and provides a promising direction for future research.

**Acknowledgement.** We would like to acknowledge the gifts from Google, Adobe, Wormpex AI, and support from NSF IIS-2047677, HDR-1934932, CCF-2019844, and IARPA WRIVA program.

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
