# MULTI-VIEW REPRESENTATION IS WHAT YOU NEED FOR POINT-CLOUD PRE-TRAINING – SUPPLEMENTARY MATERIAL

## 1 DETAILS OF FIGURE 1 IN MAIN PAPER

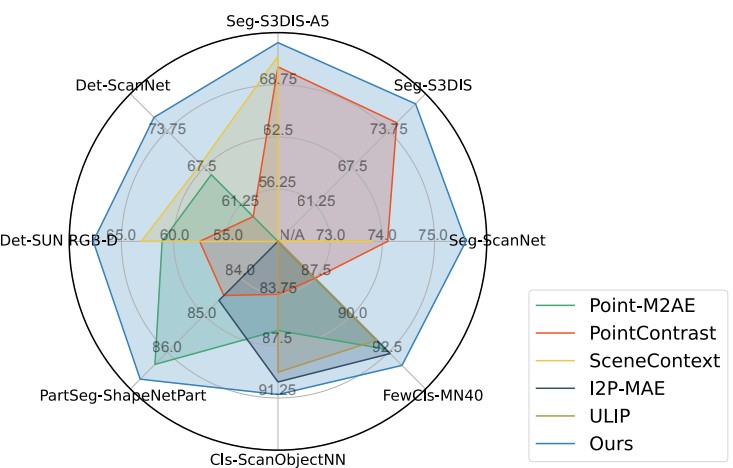

Our model (blue) achieves state-of-the-art performance across a broad range of tasks at both the scene and shape levels. The distance to the origin indicates the task result. We compared with other pre-training methods (Hou et al., 2021; Xie et al., 2020; Xue et al., 2023; Zhang et al., 2023; 2022) that achieve state-of-the-art performance on different tasks. 'Seg-ScanNet' refers to the validation mean IoU on ScanNet's semantic segmentation. 'Seg-S3DIS' indicates the mean IoU from S3DIS's 6-fold cross-validation result. 'Seg-S3DIS-A5' represents the mean IoU for the S3DIS Area 5 validation result. 'Det-ScanNet' corresponds to mAP@0.25 on the ScanNet detection task. 'Det-SUN RGB-D' signifies mAP@0.25 on the SUN RGB-D detection task. 'PartSeg-ShapeNetPart' denotes the part segmentation result on ShapeNetPart dataset. 'Cls-ScanObjectNN' denotes the shape classification result on ScanObjectnn. 'FewCls-MN40' denotes the 10-way 10-shot few shot classification result on ModelNet40.

## 2 PRE-TRAINING DETAILS

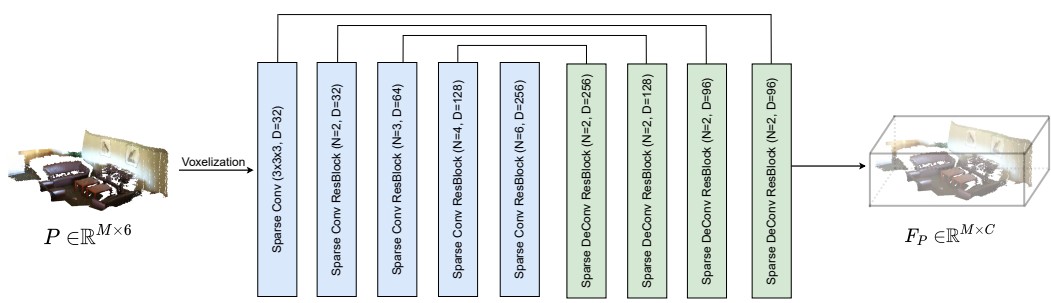

Figure 1: **Feature encoding network details.**

## 2.1 FEATURE ENCODING NETWORK DETAILS

As illustrated in Figure 5, the feature encoding network takes a colored point cloud $P \in \mathbb{R}^{M \times 6}$ as input, with the first three channels denoting coordinates and the subsequent three channels signifying RGB values. This point cloud, $P$, is then voxelized based on its 3D coordinates, which generates a grid-based representation $V \in \mathbb{R}^{M' \times 6}$, where $M'$ denotes the number of voxels.

The encoder comprises 4 Sparse Conv ResBlocks (Choy et al., 2019) with a total of 21 convolution layers, while the decoder is made up of 4 Sparse DeConv ResBlocks with 13 layers. The design of each ResBlock follows the basic 2D ResNet block pattern, and every conv/deconv layer within the network is succeeded by Batch Normalization (BN) and a ReLU activation function. The overall U-Net architecture comprises 37.85M parameters.

The network's output consists of $C$-dimensional per-voxel features $F_V \in \mathbb{R}^{M' \times C}$. Subsequently, these per-voxel features $F_V$ are interpolated to procure per-point features from the original point cloud, represented as $F_P \in \mathbb{R}^{M \times C}$. In our implementation, the value of $C$ is set at 96.

## 2.2 POINT-CLOUD FEATURE VOLUME PROJECTION DETAILS

As we implement augmentation strategies, the one-to-one correspondence between each pixel in each input view and the corresponding 3D point is established through a standard projection operation. Specifically, the relationship between a 3D point with homogeneous coordinates $P = (X, Y, Z, 1)^T$ and a 2D pixel $(x, y)^T$ is given by:

$$\frac{1}{Z'} \begin{bmatrix} x \\ y \\ 1 \end{bmatrix} = \begin{bmatrix} X' \\ Y' \\ Z' \end{bmatrix} = \begin{bmatrix} f_{s_x} & f_{s_\theta} & o_x \\ 0 & f_{s_y} & o_y \\ 0 & 0 & 1 \end{bmatrix} \begin{bmatrix} 1 & 0 & 0 & 0 \\ 0 & 1 & 0 & 0 \\ 0 & 0 & 1 & 0 \end{bmatrix} \cdot \begin{bmatrix} R & T \\ 0^T & 1 \end{bmatrix} \cdot \begin{bmatrix} X \\ Y \\ Z \\ 1 \end{bmatrix} = K \cdot I' \cdot E_i \cdot P \quad (1)$$

Here, $E_i \in E_1, E_2$ and $K$ represent the extrinsic and intrinsic matrices, respectively. $I'$ refers to the canonical projection matrix. Equation (1) enables us to derive 2D feature maps $F_1, F_2$, each having dimensions $H \times W \times C$.

## 2.3 MORE PRE-TRAINING DETAILS

During pre-training, our model comprises a total of 110M parameters and operates at 31 GFLOPs. Additionally, the procedure of converting RGB-D scans into point clouds takes roughly 10 hours on a machine equipped with a 64-core CPU. Furthermore, storing the resultant point clouds necessitates an additional 600GB of storage space.

# 3 MORE EXPERIMENT RESULTS

## 3.1 COMPARISON WITH SUPERVISED METHODS ON SEMANTIC SEGMENTATION

As shown in Table 1, compared with state-of-the-art supervised methods, our approach also achieves superior performance. Please note that our approach outperforms multi-view representation-based methods (Kundu et al., 2020; Hamdi et al., 2023) with significant improvements.

## 3.2 COMPARISON WITH PRE-TRAINING METHODS WITH DIFFERENT NETWORK BACKBONES ON SHAPE-LEVEL TASKS

Many existing pre-training approaches adopt different network backbones to in pursuit of better performance, making fair comparisons challenging. In Table 2, we compared our method against these approaches. Even under diverse settings, our method consistently achieves state-of-the-art results across all the benchmarks.

| Method | S3DIS Area 5 | | | S3DIS 6-Fold | | | ScanNet |
|---|---|---|---|---|---|---|---|
| | OA | mAcc | mIoU | OA | mAcc | mIoU | mIoU |
| PointNet (Qi et al., 2017) | 53.5 | 49.0 | 41.1 | 78.5 | 66.2 | 47.6 | - |
| PointCNN (Li et al., 2018) | 85.9 | 63.9 | 57.3 | 88.1 | 75.6 | 65.4 | 45.8 |
| SPGraph (Landrieu & Simonovsky, 2018) | 86.4 | 66.5 | 58.0 | 85.5 | 73.0 | 62.1 | - |
| PointWeb (Zhao et al., 2019) | 87.0 | 66.6 | 60.3 | 87.3 | 76.2 | 66.7 | - |
| MinkowskiNet (Choy et al., 2019) | - | 71.7 | 65.4 | - | - | - | 67.9 |
| Vision MVFusion (Kundu et al., 2020) | - | - | 65.4 | - | - | - | 74.6 |
| Voint Cloud (Hamdi et al., 2023) | - | - | 66.1 | - | - | - | 74.8 |
| PointTransformer (Zhao et al., 2021) | 90.8 | 76.5 | 70.4 | 90.2 | 81.9 | 73.5 | 70.6 |
| PointNeXt (Qian et al., 2022) | 90.7 | - | 70.8 | 90.3 | - | 74.9 | 71.5 |
| Stratified Trans (Lai et al., 2022) | 91.5 | 78.1 | 72.0 | - | - | - | 74.3 |
| PointMetaBase (Lin et al., 2022) | 91.4 | - | 72.0 | 91.3 | - | 77.0 | 72.8 |
| SR-UNet (Xie et al., 2020) | 89.1 | 75.5 | 68.2 | 90.2 | 82.1 | 73.6 | 72.2 |
| MVNet | **91.7**(91.6) | **79.5**(79.3) | **73.8**(73.3) | **91.8**(91.7) | **86.3**(86.2) | **78.3**(78.1) | **75.6**(75.2) |

Table 1: **Comparison with supervised methods on S3DIS (evaluation by 6-Fold or in Area 5) and ScanNet V2.** The average result of 3 runs is given in parentheses.

| Method | Backbone | ScanObjectNN | | | ShapeNetPart | |
|---|---|---|---|---|---|---|
| | | OBJ-BG | OBJ-ONLY | PB-T50-RS | ins. mIoU | cls. mIoU |
| Jigsaw (Sauder & Sievers, 2019) | DGCNN | 86.8 | 86.2 | 83.5 | 85.3 | - |
| OcCo (Wang et al., 2020) | DGCNN | 88.2 | 87.5 | 85.0 | - | |
| CrossPoint (Afham et al., 2022) | DGCNN | - | - | 85.5 | 83.7 | |
| ULIP (Xue et al., 2023) | Transformer | 91.3 | 89.4 | 86.4 | - | - |
| ULIP (Xue et al., 2023) | PointMLP | 93.2 | 90.4 | 89.4 | - | - |
| Point-M2AE (Zhang et al., 2022) | M2AE | 91.2 | 88.8 | 86.4 | 86.5 | 84.9 |
| I2P-MAE Zhang et al. (2023) | M2AE | 94.2 | 91.6 | 90.1 | **86.8** | **85.2** |
| MVNet-L | Transformer | **95.2** | **94.2** | **91.0** | **86.8** | **85.2** |

Table 2: **Performance comparison with pre-training approaches using different network backbones on shape-level downstream tasks.**

## 3.3 FEW-SHOT CLASSIFICATION ON MODELNET40

We conduct few-shot classification on ModelNet40, adopting the "K-way N-shot" configurations detailed in previous studies (Wang et al., 2021; Yu et al., 2022; Pang et al., 2022). Specifically, from the 40 available classes, we randomly select K and then sample N+20 3D shapes from each class. Of these, N shapes are designated for training, while 20 are reserved for testing. We assess MVNet's performance across four distinct few-shot scenarios: 5-way 10-shot, 5-way 20-shot, 10-way 10-shot, and 10-way 20-shot. To counteract potential biases from random sampling, we carry out 10 separate runs for each scenario, subsequently reporting both the average accuracy and standard deviation. Remarkably, our model excels using the standard transformer architecture.

## 3.4 LABEL EFFICIENCY TRAINING

Pre-training enables models to be effectively fine-tuned with a minimal quantity of labeled data. In our study, we examine the label efficiency of our model in 3D object detection by adjusting the proportion of supervised training data used. The results of this study can be viewed in Figure 2.

We utilize VoteNet (Qi et al., 2019) as the training network, incorporating 20%, 40%, 60%, and 80% of the training data derived from the ScanNet and SUN RGB-D datasets. Interestingly, we observe that our pre-training approach yields larger improvements when less labeled data is used. Remarkably, with approximately 60% of training data from ScanNet/SUN RGB-D, our model performs comparably to a model trained from scratch with full data.

To further prove the efficacy of our method, We also compare our method with the most recent works (Xie et al., 2020; Chen et al., 2022) on SUN RGB-D detection task. During these experiments, we trained our model using varied proportions of the dataset, specifically 20%, 50%, and 100%, and subsequently evaluated the performance on the same test dataset. We reported the mAP @0.5 results in Table 4. Our observations indicate that our pre-training method yields more significant gains with reduced amounts of labeled training data. Notably, with only 50% of the training data, our method surpasses the performance achieved by training from scratch using the full 100% dataset.

| Method | 5-way | | 10-way | |
|---|---|---|---|---|
| | 10-shot | 20-shot | 10-shot | 20-shot |
| DGCNN[†] | $31.6 \pm 2.8$ | $40.8 \pm 4.6$ | $19.9 \pm 2.1$ | $16.9 \pm 1.5$ |
| OcCo | $90.6 \pm 2.8$ | $92.5 \pm 1.9$ | $82.9 \pm 1.3$ | $86.5 \pm 2.2$ |
| CrossPoint | $92.5 \pm 3.0$ | $94.9 \pm 2.1$ | $83.6 \pm 5.3$ | $87.9 \pm 4.2$ |
| Transformer[†] | $87.8 \pm 5.2$ | $93.3 \pm 4.3$ | $84.6 \pm 5.5$ | $89.4 \pm 6.3$ |
| OcCo | $94.0 \pm 3.6$ | $95.9 \pm 2.3$ | $89.4 \pm 5.1$ | $92.4 \pm 4.6$ |
| PointBERT | $94.6 \pm 3.1$ | $96.3 \pm 2.7$ | $91.0 \pm 5.4$ | $92.7 \pm 5.1$ |
| MaskDiscr | $95.0 \pm 3.7$ | $97.2 \pm 1.7$ | $91.4 \pm 4.0$ | $93.4 \pm 3.5$ |
| PointMAE | $96.3 \pm 2.5$ | $97.8 \pm 1.8$ | $92.6 \pm 4.1$ | $95.0 \pm 3.0$ |
| MVNet | $\mathbf{97.2 \pm 1.8}$ | $\mathbf{98.3 \pm 1.8}$ | $\mathbf{93.4 \pm 3.4}$ | $\mathbf{95.8 \pm 3.2}$ |

Table 3: **Few-shot classification on ModelNet40.** We report the average accuracy (%) and standard deviation (%) of 10 independent experiments. [†] represents the *from scratch* results and all other methods represent the *fine-tuning* results using pretrained weights.

| Method | 20% | 50% | 100% |
|---|---|---|---|
| Train from scratch | 18.8 | 24.7 | 32.9 |
| PointContrast | 24.5 | 29.2 | 37.5 |
| 4DContrast | 26.3 | 31.5 | 38.2 |
| MVNet | **28.5** | **33.2** | **39.6** |

Table 4: **Label efficient training on SUN RGB-D object detection task.** We take the VoteNet as network architecture and compare our method with PointContrast and 4DContrast.

These findings suggest that our pre-training approach can enhance the performance of downstream tasks even when less data is available, thereby increasing the efficiency of the training process.

## 4 VISUALIZATION OF MULTI-VIEW CONSISTENCY MODULE PREDICTION

Figure 3 shows the qualitative results of the multi-view consistency module prediction. The figure consists of various elements, each showcasing two images from distinct viewpoints. This visualization emphasizes the accuracy and efficiency of our multi-view consistency module. Furthermore, the results demonstrate that our pre-trained model learns the ability to capture 3d features.

## 5 QUALITATIVE RESULTS ON SEMANTIC SEGMENTATION TASK

In this section, we show more qualitative results on the S3DIS Area 5 semantic segmentation task. We compare our method with SceneContext (Hou et al., 2021), the second-best approach.

## 6 BROADER IMPACTS

The total emission is estimated to be 161.28 kgCO$_2$eq, equivalent to 652 km driven by an average car. This emission estimation is conducted using the Machine Learning Impact calculator presented in (Lacoste et al., 2019). To mitigate repetitive labor and negative environmental impact in future research, we will release our open-source implementation together with trained network weights after the anonymous period.

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

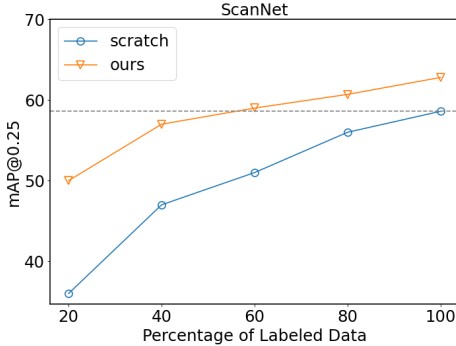 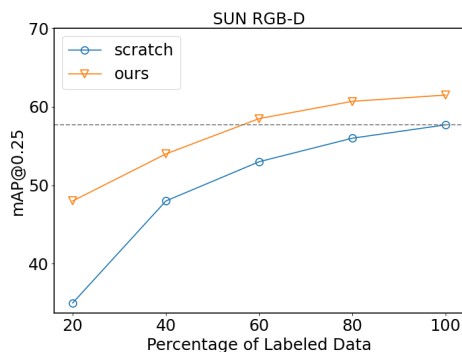

Figure 2: **Illustration of Label efficiency training.** Our model is pre-trained on ScanNet and subsequently fine-tuned separately on both ScanNet and SUN RGBD. We take VoteNet as network architecture. During this fine-tuning process, we employ varying percentages of labeled training data. Demonstrating its efficacy, our pre-training model not only surpasses the performance of models trained from scratch but also achieves comparable results utilizing only 60% of the labeled data.

*Conference, Tel Aviv, Israel, October 23–27, 2022, Proceedings, Part XXXII*, pp. 543–560. Springer, 2022.

Christopher Choy, JunYoung Gwak, and Silvio Savarese. 4d spatio-temporal convnets: Minkowski convolutional neural networks. In *Proceedings of the IEEE/CVF conference on computer vision and pattern recognition*, pp. 3075–3084, 2019.

Abdullah Hamdi, Silvio Giancola, and Bernard Ghanem. Voint cloud: Multi-view point cloud representation for 3d understanding. In *The Eleventh International Conference on Learning Representations, ICLR 2023, Kigali, Rwanda, May 1-5, 2023*. OpenReview.net, 2023. URL `https://openreview.net/pdf?id=IpGgfpMucHj`.

Ji Hou, Benjamin Graham, Matthias Nießner, and Saining Xie. Exploring data-efficient 3d scene understanding with contrastive scene contexts. In *Proceedings of the IEEE/CVF Conference on Computer Vision and Pattern Recognition*, pp. 15587–15597, 2021.

Abhijit Kundu, Xiaoqi Yin, Alireza Fathi, David A. Ross, Brian Brewington, Thomas A. Funkhouser, and Caroline Pantofaru. Virtual multi-view fusion for 3d semantic segmentation. In Andrea Vedaldi, Horst Bischof, Thomas Brox, and Jan-Michael Frahm (eds.), *Computer Vision - ECCV 2020 - 16th European Conference, Glasgow, UK, August 23-28, 2020, Proceedings, Part XXIV*, volume 12369 of *Lecture Notes in Computer Science*, pp. 518–535. Springer, 2020. doi: 10.1007/978-3-030-58586-0\_31. URL `https://doi.org/10.1007/978-3-030-58586-0_31`.

Alexandre Lacoste, Alexandra Luccioni, Victor Schmidt, and Thomas Dandres. Quantifying the carbon emissions of machine learning. *arXiv preprint arXiv:1910.09700*, 2019.

Xin Lai, Jianhui Liu, Li Jiang, Liwei Wang, Hengshuang Zhao, Shu Liu, Xiaojuan Qi, and Jiaya Jia. Stratified transformer for 3d point cloud segmentation. In *Proceedings of the IEEE/CVF Conference on Computer Vision and Pattern Recognition*, pp. 8500–8509, 2022.

Loic Landrieu and Martin Simonovsky. Large-scale point cloud semantic segmentation with super-point graphs. In *Proceedings of the IEEE conference on computer vision and pattern recognition*, pp. 4558–4567, 2018.

Yangyan Li, Rui Bu, Mingchao Sun, Wei Wu, Xinhan Di, and Baoquan Chen. Pointcnn: Convolution on x-transformed points. *Advances in neural information processing systems*, 31:820–830, 2018.

Haojia Lin, Xiawu Zheng, Lijiang Li, Fei Chao, Shanshan Wang, Yan Wang, Yonghong Tian, and Rongrong Ji. Meta architecure for point cloud analysis. *arXiv preprint arXiv:2211.14462*, 2022.

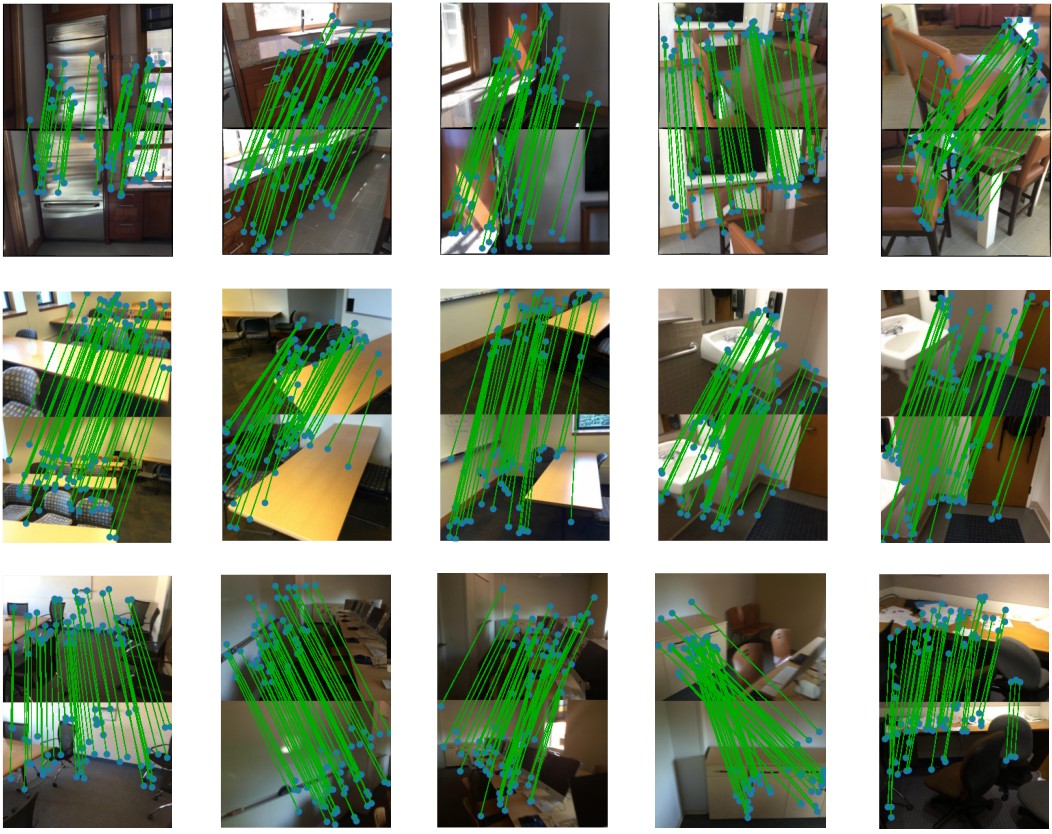

Figure 3: **Qualitative results of multi-view consistency module prediction.** Each element consists of two different view images. The query points and prediction points are visualized using the blue dot, and they are connected by the green line.

Yatian Pang, Wenxiao Wang, Francis EH Tay, Wei Liu, Yonghong Tian, and Li Yuan. Masked autoencoders for point cloud self-supervised learning. In *Computer Vision–ECCV 2022: 17th European Conference, Tel Aviv, Israel, October 23–27, 2022, Proceedings, Part II*, pp. 604–621. Springer, 2022.

Charles R Qi, Hao Su, Kaichun Mo, and Leonidas J Guibas. Pointnet: Deep learning on point sets for 3d classification and segmentation. In *Proceedings of the IEEE conference on computer vision and pattern recognition*, pp. 652–660, 2017.

Charles R Qi, Or Litany, Kaiming He, and Leonidas J Guibas. Deep hough voting for 3d object detection in point clouds. In *proceedings of the IEEE/CVF International Conference on Computer Vision*, pp. 9277–9286, 2019.

Guocheng Qian, Yuchen Li, Houwen Peng, Jinjie Mai, Hasan Hammoud, Mohamed Elhoseiny, and Bernard Ghanem. Pointnext: Revisiting pointnet++ with improved training and scaling strategies. *Advances in Neural Information Processing Systems*, 35:23192–23204, 2022.

Jonathan Sauder and Bjarne Sievers. Self-supervised deep learning on point clouds by reconstructing space. *arXiv preprint arXiv:1901.08396*, 2019.

Hanchen Wang, Qi Liu, Xiangyu Yue, Joan Lasenby, and Matt J Kusner. Unsupervised point cloud pre-training via occlusion completion. In *Proceedings of the IEEE/CVF international conference on computer vision*, pp. 9782–9792, 2021.

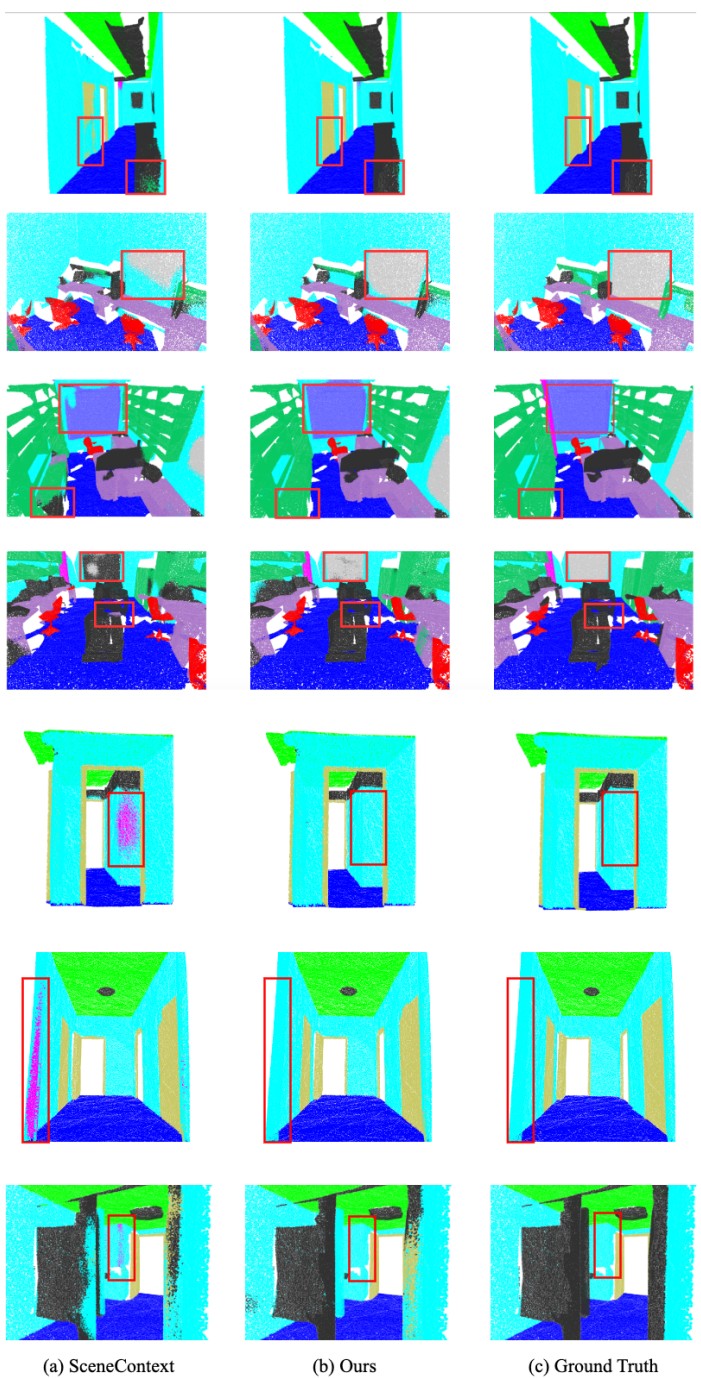

Figure 4: **Qualitative results on S3DIS semantic segmentation task.** We compare our method with the second-best approach, SceneContext (Hou et al., 2021).

Peng-Shuai Wang, Yu-Qi Yang, Qian-Fang Zou, Zhirong Wu, Yang Liu, and Xin Tong. Unsupervised 3d learning for shape analysis via multiresolution instance discrimination. *ACM Trans. Graphic*, 2020.

Saining Xie, Jiatao Gu, Demi Guo, Charles R Qi, Leonidas Guibas, and Or Litany. Pointcontrast: Unsupervised pre-training for 3d point cloud understanding. In *European Conference on Computer Vision*, pp. 574–591. Springer, 2020.

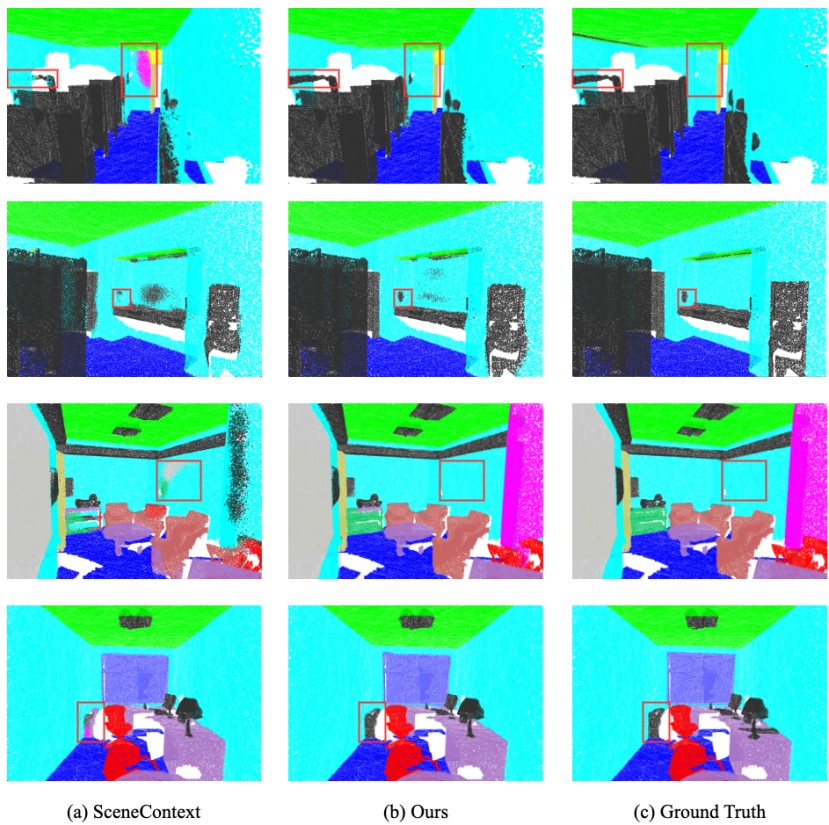

(a) SceneContext     (b) Ours     (c) Ground Truth

Figure 5: **More qualitative results on S3DIS semantic segmentation task.** We compare our method with the second-best approach, SceneContext (Hou et al., 2021).

Le Xue, Mingfei Gao, Chen Xing, Roberto Martín-Martín, Jiajun Wu, Caiming Xiong, Ran Xu, Juan Carlos Niebles, and Silvio Savarese. Ulip: Learning a unified representation of language, images, and point clouds for 3d understanding. In *Proceedings of the IEEE/CVF Conference on Computer Vision and Pattern Recognition*, pp. 1179–1189, 2023.

Xumin Yu, Lulu Tang, Yongming Rao, Tiejun Huang, Jie Zhou, and Jiwen Lu. Point-bert: Pre-training 3d point cloud transformers with masked point modeling. In *Proceedings of the IEEE/CVF Conference on Computer Vision and Pattern Recognition*, pp. 19313–19322, 2022.

Renrui Zhang, Ziyu Guo, Peng Gao, Rongyao Fang, Bin Zhao, Dong Wang, Yu Qiao, and Hongsheng Li. Point-m2ae: multi-scale masked autoencoders for hierarchical point cloud pre-training. *arXiv preprint arXiv:2205.14401*, 2022.

Renrui Zhang, Liuhui Wang, Yu Qiao, Peng Gao, and Hongsheng Li. Learning 3d representations from 2d pre-trained models via image-to-point masked autoencoders. In *Proceedings of the IEEE/CVF Conference on Computer Vision and Pattern Recognition*, pp. 21769–21780, 2023.

Hengshuang Zhao, Li Jiang, Chi-Wing Fu, and Jiaya Jia. Pointweb: Enhancing local neighborhood features for point cloud processing. In *Proceedings of the IEEE/CVF conference on computer vision and pattern recognition*, pp. 5565–5573, 2019.

Hengshuang Zhao, Li Jiang, Jiaya Jia, Philip HS Torr, and Vladlen Koltun. Point transformer. In *Proceedings of the IEEE/CVF International Conference on Computer Vision*, pp. 16259–16268, 2021.