# OpenReview forum: "Multi-View Representation is What You Need for Point-Cloud Pre-Training"
_ICLR.cc/2024/Conference — ICLR 2024 poster_

### Official Review · Reviewer_1Vvn · 2023-10-27

**Soundness:** 2 fair
**Presentation:** 3 good
**Contribution:** 2 fair
**Rating:** 6
**Confidence:** 4

**Summary:**

The paper proposes a pre-training method for 3D point clouds that leverages 2D pre-trained foundation models. The method performs multiview consistency pre-training with a distillation loss from 2D pre-trained model. The experiments show the effectiveness of this method on several tasks on point cloud data.

**Strengths:**

The major strength of the paper is the effectiveness and simplicity of the method. Although I think the paper does not have a big technical contribution, since knowledge distillation from 2D foundation models is becoming a common approach, I like the paper since it is simpler than other concurrent works [1] and shows significant improvements.

[1] Bridging the Domain Gap: Self-Supervised 3D Scene Understanding with Foundation Models, Chen and Li 2023

**Weaknesses:**

Although I like the concept, I think the evaluation should be improved. In particular, I am concerned about the weight initialization scheme used. The paper states that only the weights of the encoder are re-used for initialization since it leads to better performance. However, all previous approaches have used the pre-trained weights from the encoder and decoder. Therefore, it is difficult to assess if this pre-training setup improves over previous work. Since this work uses the same backbone as previous pre-training works and these methods provide pre-trained weights, it would be easy and necessary to provide a comparison of fine-tuned models for previous works where only the encoder is reused. Moreover, numbers for a fine-tuned model with the proposed method where the decoder weights are also reused should be provided. With this experiment, we will be able to assess if this is only necessary for the proposed pre-training method or if all pre-training strategies suffer from it. More importantly, we will be able to assess if the described improvement on the paper really comes from the pre-training strategy or from the fine-tuning setup used. I would rate the paper as marginally above acceptance and I would change my rating after rebuttal when these numbers are available.

Moreover, it is not clear the masking ratio ablation experiment. There is no description in the methods section of any masking applied during pre-training, so I think it will be necessary to clarify how this masking is applied during pre-training.

**Questions:**

See weaknesses.

---

> ### Author Response · Authors · 2023-11-20
>
> We sincerely appreciate the reviewer's insightful feedback and positive comments. In response to the concerns raised, we offer the following clarifications and additional information.
>
> **Q: All previous approaches have used the pre-trained weights from the encoder and decoder.**
>
> We thank the reviewer for raising this question. We would like to clarify this question here for different downstream tasks.
>
> In the shape classification task, only the encoder is employed across all pre-training methods. This is because the classification task does not necessitate a decoder.
>
> Similarly, for the object detection task, pre-training methods exclusively utilize the encoder. The detection decoder is typically custom-designed and differs significantly from those used in pre-training.
>
> Regarding the semantic segmentation task, while the decoder could theoretically be reused, some recent pre-training methods like DepthContrast[1] do not incorporate a decoder during pre-training. Consequently, these methods do not offer a decoder component that can be initialized for downstream tasks.
>
> In our experimental analysis, we observed that the encoder weights play a pivotal role. As demonstrated in the table below, when transferring pre-trained decoder weights, our model consistently outperforms the second-best method on semantic segmentation tasks.
>
> | Method              | Backbone | S3DIS Area 5 mIoU | S3DIS 6-Fold mIoU | ScanNet mIoU |
> | ------------------- | -------- | ----------------- | ----------------- | ------------ |
> | Train from scratch  | SR-UNet  | 68.2              | 73.6              | 72.2         |
> | SceneContext        | SR-UNet  | 72.2              | -                 | 73.8         |
> | Ours (encoder only)        | SR-UNet  | **73.8**          | **78.3**          | **75.6**     |
> | Ours (encoder and decoder) | SR-UNet  | 73.5              | 78.0              | 75.4         |
>
> **Q: Masking ratio ablation experiment.**
>
> We express our gratitude to the reviewer for the meticulous attention in pointing this out. We intend to provide a more detailed explanation in our revised manuscript. Our method involved masking the input point cloud guided by the corresponding RGB image. Following the conventional practice in masked autoencoder-based pre-training [2], we divided the image into regular non-overlapping patches. Then we sample a subset of patches and mask (i.e., remove) the remaining ones. The remaining part is projected onto the point cloud, forming the input for our method. We employ this masking operation as a data augmentation technique, which assists in the development of more robust features during pre-training. As evidenced in Table 4(b) (referenced below), an appropriate masking operation during pre-training could enhance performance in downstream tasks.
>
> | Mask Ratio (%) | ScanNet  |
> | -------------- | -------- |
> | 10             | 74.8     |
> | 30             | **75.6** |
> | 60             | 74.2     |
> | 80             | 73.5     |
>
> [1] Zhang, Zaiwei, et al. "Self-supervised pretraining of 3d features on any point-cloud." In ICCV 2021.
>
> [2] He, Kaiming, et al. "Masked autoencoders are scalable vision learners." In CVPR 2022.

---

> > ### Comment · Reviewer_1Vvn · 2023-11-22
> > **Post-rebuttal assessment**
> >
> > I would like to thank the authors for the additional experiments. All my concerns have been addressed and therefore I will keep my positive rating.

---

> > > ### Author Response · Authors · 2023-11-22
> > > **Thank you for your positive rating!**
> > >
> > > Thank you for the positive rating! We are pleased to know that our responses have successfully addressed your concerns. We also extend our sincere gratitude for the time and effort you have invested in reviewing our work. Your insights are greatly appreciated.

---

### Official Review · Reviewer_JZVU · 2023-10-29

**Soundness:** 3 good
**Presentation:** 3 good
**Contribution:** 3 good
**Rating:** 6
**Confidence:** 5

**Summary:**

This paper proposes to leverage the pre-trained 2D **networks** instead of data for 3D point cloud pretraining. Specifically, a 3D network is first applied to extract 3D feature volumes that are then projected into image-level embeddings with the help of depth information. Next, the key idea is to make the projected image features close to their counterparts that are pretrained using powerful 2D vision foundation models such as Dinov2, CLIP, SAM. Additionally, to realize multi-view consistency, another module is adapted to maintain the point correspondences across different views, yielding the whole proposed method called MVNet. For verifying the effectiveness of MVNet, experiments are conducted on various benchmarks and tasks.

**Strengths:**

1.	The paper is well-organized, well-written, and easy to follow.
2.	The key idea to leverage the pre-trained 2D networks and learn the 3D-2D correspondence at the feature level is interesting and effective. Such pretext task design provides a new direction for future research on 3D pretraining.
3.	The proposed method achieves **superior results** on various benchmarks and tasks.
4.	The supplementary material provides more experimental results and states the broader impacts. Additionally, the qualitative results of multi-view consistency predictions further prove the effectiveness of the proposed method.
5.	Some possible limitations and future potentials are also discussed.

**Weaknesses:**

1.	The **masking ratio** is *only discussed* in the ablation study, which is very confusing. Some more details about this masking operation should be discussed in the method section. For example, why do you use this operation? Why can it benefit the proposed method?
2.	There are several **important recent related works** [1, 2] that should be discussed in Related Works and compared in the tables in the main paper.
3.	The proposed method introduces two modules, thus the **#params and FLOPs** during pretraining (all the parts that need to be trained online in Fig. 2 should be counted) need to be provided and compared with other methods to show the efficiency of the proposed method.
4.	The proposed method requires the **pre-processing** on RGB-D scans to generate point clouds. This operation may bring extra processing time and disk usage. How long will it take? How much extra storage will this need? This should be at least discussed in the supplementary material.
Also, this may be a limitation of the proposed method, which should be mentioned in the limitation section.
5.	Some **qualitative/visualization results** of different tasks such as 3D segmentation, and detection should be provided in the supplementary material.

**Refs**:
[1] Mask3D: Pre-training 2D Vision Transformers by Learning Masked 3D Priors. CVPR 2023.
[2] MM-3DScene: 3D Scene Understanding by Customizing Masked Modeling with Informative-Preserved Reconstruction and Self-Distilled Consistency. CVPR 2023.

**Questions:**

This work shows great potential to serve as a new pretraining method for 3D point cloud processing. Hope the code and the pretrained models could be public upon the acceptance of this paper.

The major concerns are detailed in the weaknesses part. The missing discussions, experimental results and qualitative results are expected to be provided during the rebuttal.

---

> ### Author Response · Authors · 2023-11-21
>
> We are deeply grateful to the reviewer for their insightful feedback and positive comments. In addressing the concerns raised, we offer the following clarifications and additional information.
>
> **Q: Masking ratio ablation experiment.**
>
> We express our gratitude to the reviewer for the meticulous attention in pointing this out. We intend to provide a more detailed explanation in our revised manuscript. Our method involved masking the input point cloud guided by the corresponding RGB image. Following the conventional practice in masked autoencoder-based pre-training [1], we divided the image into regular non-overlapping patches. Then we sample a subset of patches and mask (i.e., remove) the remaining ones. The remaining part is projected onto the point cloud, forming the input for our method. We employ this masking operation as a data augmentation technique, which assists in the development of more robust features during pre-training. As evidenced in Table 4(b) (referenced below), an appropriate masking operation during pre-training could enhance performance in downstream tasks.
>
> | Mask Ratio (%) | ScanNet  |
> | -------------- | -------- |
> | 10             | 74.8     |
> | 30             | **75.6** |
> | 60             | 74.2     |
> | 80             | 73.5     |
>
> **Q: Important recent related works.**
>
> We are thankful to the reviewer for bringing these related works to our attention. We will include a comprehensive discussion of these works in our revised manuscript. Mask3D introduces a method of pre-training in 2D by harnessing 3D priors, with the aim of enhancing 2D task performance. Our approach, conversely, seeks to apply 2D knowledge to augment 3D tasks, establishing a reversed transfer of knowledge. Given the differing objectives, a direct comparison with Mask3D is not straightforward.
>
> MM-3DScene has proposed an innovative masking model for point cloud pre-training. We have conducted a comparative analysis with their method on the object detection task, and the results are presented in the table that follows. Our method outperforms MM-3DScene in most cases.
>
> | Method             | Architecture | ScanNet mAP @0.25 | ScanNet mAP @0.5 | SUN RGB-D mAP @0.25 | SUN RGB-D mAP @0.5 |
> | ------------------ | ------------ | ---------------- | --------------- | ------------------ | ----------------- |
> | Train from scratch | VoteNet      | 58.7             | 35.4            | 57.7               | 32.9              |
> | MM-3DScene         | VoteNet      | 63.1             | **41.5**        | 60.6               | 37.3              |
> | MVNet              | VoteNet      | **64.0**         | **41.5**        | **62.0**           | **39.6**          |
>
>
> We also compare our method with MM-3DScene on the semantic segmentation task. Since MM-3DScene employs Point Transformer as the encoder backbone, we ensured a fair comparison by implementing Point Transformer as the backbone during our method's pre-training phase. The results, which are presented in the subsequent table, demonstrate that our method outperforms MM-3DScene under the same experimental setting.
>
> | Method             | Backbone          | ScanNet mIoU | S3DIS Area5 mIoU |
> | ------------------ | ----------------- | ------------ | ---------------- |
> | Train from scratch | Point Transformer | 70.6         | 70.4             |
> | MM-3DScene         | Point Transformer | 72.8         | 71.9             |
> | MVNet              | Point Transformer | **75.2**     | **73.5**         |
>
> **Q: The number of params and FLOPS.**
>
> We are grateful to the reviewer for highlighting this matter. During pre-training, our model comprises a total of 110M parameters and operates at 31 GFLOPs. As a comparison, a preivous state-of-the-art method, 4DContrast, contains 71M parameters. It is important to note that while contrastive learning-based methods typically necessitate large batch sizes, our method does not require this. Consequently, our approach offers greater flexibility across various hardware configurations.
>
> **Q: Pre-processing discussion.**
>
> We extend our gratitude to the reviewers for bringing this to our attention. We acknowledge the significance of this point and will address it, adding a corresponding note to the limitations section in our revised manuscript. The process of projecting RGB-D scans to point clouds requires approximately 10 hours on a 64-core CPU machine. Storaging the generated point cloud would require  ~600G extra space.
>
> **Q: Qualitative/visualization results.**
>
> We are thankful to the reviewer for the suggestion. In response, we have included additional qualitative results for the semantic segmentation task in the updated supplementary material.
>
> [1] He, Kaiming, et al. "Masked autoencoders are scalable vision learners." In CVPR 2022.

---

> ### Comment · Reviewer_JZVU · 2023-11-22
>
> I appreciate the detailed response from the authors.
>
> Most of the responses addressed my concerns, so I will keep my score of 6.
>
> Please also note that, the qualitative results shown in the supplementary material are still not enough, more results are expected to be provided in the final version.
>
> To summarize, I lean to accept this paper under the condition of **providing more qualitative results and adding the missing discussions of related works and experimental results listed in the rebuttal**.

---

> > ### Author Response · Authors · 2023-11-22
> > **Thank you for your response!**
> >
> > Thank you for your insightful and positive feedback! In line with the reviewer's suggestions, we have added more qualitative results in the supplementary material. We have also added the missing discussions of related works and the updated experimental results in the main paper. We look forward to further discussions should you have any follow-up concerns!

---

### Official Review · Reviewer_GBMb · 2023-10-31

**Soundness:** 2 fair
**Presentation:** 3 good
**Contribution:** 2 fair
**Rating:** 6
**Confidence:** 4

**Summary:**

This paper introduces a 3D representation learning approach that involves projecting 3D dense features into 2D images. This method employs multi-level feature maps for cross-modal knowledge distillation and employs coordinate mapping as a pretext task for achieving multi-view consistency learning. Extensive experiments demonstrate the effectiveness of both modules in various downstream tasks, including 3D shape classification, part segmentation, 3D object detection, and semantic segmentation.

**Strengths:**

1. The proposed method exhibits a high degree of applicability, yielding favorable results across indoor scene-level datasets such as ScanNet and S3DIS, as well as at the object level dataset such as ScanObjectNN.

2. The approach of projecting 3D features to 2D and learning consistency through point mapping aligns with physical intuition.

3. The paper is well-written and ablation experiments are well-designed.

**Weaknesses:**

1. Given that the model can generate dense features containing rich semantics, I strongly recommend incorporating zero-shot experiments, including zero-shot classification experiments on datasets such as ModelNet and LVIS, as well as zero-shot semantic segmentation experiments on ScanNet and S3DIS.

2. The classification experiments at the object level lack references and comparisons to recent works, such as ULIP-2 [Liu et al., 2023] and ReCon [Qi et al., 2023]. Furthermore, MVNet is trained on Objaverse and includes multiple versions with different parameters, making the comparisons in Table 3 unfair in terms of both the training dataset and model parameters.

3. Some papers are repeatedly cited in the bib, such as: "Charles R. Qi, Or Litany, Kaiming He, and Leonidas J. Guibas. Deep hough voting for 3d object detection in point clouds. In Proceedings of the IEEE/CVF International Conference on Computer Vision (ICCV), October 2019a."

4. The result of ScanObjectNN in Figure 1 is different from the result in Table 3.

[Qi et al., 2023] Contrast with Reconstruct: Contrastive 3D Representation Learning Guided by Generative Pretraining. In ICML.

[Liu et al., 2023] ULIP-2: Towards Scalable Multimodal Pre-training for 3D Understanding. arXiv preprint.

**Questions:**

1. Why not directly use the transformation matrix of two views as training targets within the multi-view consistency module?

2. Is voting strategy employed in the classification of object point clouds?

3. I am curious to understand why SAM yields inferior performance as a 2D encoder compared to DINOv2.

---

> ### Author Response · Authors · 2023-11-21
> **Official Comment by Authors (Part 1)**
>
> We extend our heartfelt thanks to the reviewer for the positive feedback. In addressing the concerns raised, we are pleased to provide the following clarifications and additional information.
>
> **Q: Zero-shot experiments.**
>
> We extend our thanks to the reviewer for highlighting this aspect. Our method, not being designed to align learned features with the language domain, faces challenges in direct application to zero-shot experiments like PointCLIP [1] or ULIP [2]. Nevertheless, to demonstrate the effectiveness of our approach, we have included results from few-shot learning experiments conducted on ModelNet40. These results can be found in Table 3 of the Supplementary Material. The table below also presents additional results pre-trained on ShapeNet for a comprehensive view.
>
> | Model        | Pre-train dataset | 5-way 10-shot  | 5-way 20-shot  | 10-way 10-shot | 10-way 20-shot |
> | ------------ | -------------- | -------------- | -------------- | -------------- | -------------- |
> | Transformer† | - | 87.8 ± 5.2     | 93.3 ± 4.3     | 84.6 ± 5.5     | 89.4 ± 6.3     |
> | OcCo         | ShapeNet | 94.0 ± 3.6     | 95.9 ± 2.3     | 89.4 ± 5.1     | 92.4 ± 4.6     |
> | PointBERT    | ShapeNet | 94.6 ± 3.1     | 96.3 ± 2.7     | 91.0 ± 5.4     | 92.7 ± 5.1     |
> | MaskDiscr    | ShapeNet | 95.0 ± 3.7     | 97.2 ± 1.7     | 91.4 ± 4.0     | 93.4 ± 3.5     |
> | PointMAE     | ShapeNet | 96.3 ± 2.5     | 97.8 ± 1.8     | 92.6 ± 4.1     | 95.0 ± 3.0     |
> | MVNet-B       | ShapeNet | 96.9 ± 1.9 | 98.1 ± 1.7 | 93.1 ± 3.5 | 95.4 ± 3.4 |
> | MVNet-B       | Objaverse | 97.2 ± 1.8 | 98.3 ± 1.8 | 93.4 ± 3.4 | 95.8 ± 3.2 |
>
> To further prove the efficacy of our method, we conducted the label-efficient learning experiments on the scene-level SUN RGB-D object detection task. During these experiments, we trained our model using varied proportions of the dataset, specifically 20%, 50%, and 100%, and subsequently evaluated the performance on the same test dataset. We reported the mAP @0.5 results in the table below. Our observations indicate that our pre-training method yields more significant gains with reduced amounts of labeled training data. Notably, with only 50% of the training data, our method surpasses the performance achieved by training from scratch using the full 100% dataset.
>
> | Method             | 20%      | 50%      | 100%     |
> | ------------------ | -------- | -------- | -------- |
> | Train from scratch | 18.8     | 24.7     | 32.9     |
> | PointContrast      | 24.5     | 29.2     | 37.5     |
> | 4DContrast         | 26.3     | 31.5     | 38.2     |
> | MVNet              | **28.5** | **33.2** | **39.6** |
>
> We hope that these relevant experiments will effectively address the reviewer's concerns.
>
> [1] Zhang, Renrui, et al. "Pointclip: Point cloud understanding by clip." In CVPR 2022.
>
> [2] Xue, Le, et al. "ULIP: Learning a unified representation of language, images, and point clouds for 3D understanding." In CVPR 2023.

---

> > ### Author Response · Authors · 2023-11-21
> > **Official Comment by Authors (Part 2)**
> >
> > **Q: The classification experiments at the object level lack references and comparisons to recent works.**
> >
> > We thank the reviewer for highlighting this aspect. ULIP-2 represents a more advanced version of ULIP. It aligns multimodal features across 3d shapes, 2d images and language description. However, its focus on aligning only the holistic features of 3D shapes and conducting experiments solely on the 3D shape classification task somewhat restricts its applicability. In contrast, our method demonstrates greater adaptability and generality, effectively applicable to different shape-level tasks such as part segmentation and even scene-level tasks. Additionally, as ULIP-2 has not yet been published, we consider it a concurrent relevant work. We will include a discussion about it in our revised manuscript.
> >
> > To further validate our approach, we compared our method with ULIP-2 in the following table. We first pre-trained our model on ShapeNet and show the ScanObjectNN PB-T50-RS classification task results. Our model shows the same result as ULIP-2 under the same experimental setting.
> >
> > | Method             | Backbone    | Pre-train dataset | ScanObjectNN PB-T50-RS |
> > | ------------------ | ----------- | ----------------- | ---------------------- |
> > | Train from scratch | Transformer | -                 | 77.2                   |
> > | Point-MAE          | Transformer | ShapeNet          | 85.2                   |
> > | ULIP               | Transformer | ShapeNet          | 86.4                   |
> > | ULIP-2             | Transformer | ShapeNet          | 86.7                   |
> > | MVNet-B            | Transformer | ShapeNet          | 86.7                   |
> >
> > We have also included experimental results from pre-training on the Objaverse dataset in the table below. Our findings reveal that when using a smaller backbone, our model underperforms compared to ULIP and ULIP-2. We attribute this to ULIP and ULIP-2's pre-training strategies, which are seemingly more effective at capturing the global features of 3D shapes. However, as we scale up the backbone size, our model demonstrates significantly improved performance. This highlights the robust scalability of our method, showcasing its enhanced effectiveness with larger backbones.
> >
> > | Method             | Backbone      | Pre-train dataset | ScanObjectNN PB-T50-RS |
> > | ------------------ | ------------- | ----------------- | ---------------------- |
> > | Train from scratch | Transformer   | -                 | 77.2                   |
> > | Point-MAE          | Transformer   | Objaverse         | 84.9                   |
> > | ULIP               | Transformer   | Objaverse         | 88.7                   |
> > | ULIP-2             | Transformer   | Objaverse         | **89.0**               |
> > | MVNet-B            | Transformer   | Objaverse         | 87.8                   |
> > | ULIP-L             | Transformer-L | Objaverse         | 89.1                   |
> > | ULIP-2-L           | Transformer-L | Objaverse         | 90.5                   |
> > | MVNet-L            | Transformer-L | Objaverse         | **91.0**               |
> >
> > We also compared our mehod with ULIP-2 on the ShapeNetPart segmentation task in the table below. The results demonstrate that our method significantly outperforms both ULIP and ULIP-2. This superior performance underscores the ability of our model to learn richer and more fine-grained features during the pre-training phase.
> >
> > | Method             | Backbone    | Pre-train dataset | ShapeNetPart cls. mIoU |
> > | ------------------ | ----------- | ----------------- | ---------------------- |
> > | Train from scratch | Transformer | -                 | 83.4                   |
> > | ULIP               | Transformer | ShapeNet          | 84.2                   |
> > | ULIP-2             | Transformer | ShapeNet          | 84.4                   |
> > | MVNet              | Transformer | ShapeNet          | **84.8**               |
> >
> >
> >
> > ReCon introduces an innovative pre-training approach that combines generative and contrastive learning, along with a novel Recon-block tailored for transformer architecture. While it demonstrates promising results in classification tasks, its application is primarily confined to shape-level analysis. To make a fair comparison, we also pre-trained our model with their proposed ReCon-Block on the ShapeNet dataset and show the ScanObjectNN PB-T50-RS classification results without voting strategy below. Our method outperforms ReCon significantly.
> >
> > | Method             | Backbone          | Pre-train dataset | ScanObjectNN PB-T50-RS |
> > | ------------------ | ----------------- | ----------------- | ---------------------- |
> > | Train from scratch | ReCon Transformer | -                 | 81.6                   |
> > | ReCon              | ReCon Transformer | ShapeNet          | 89.7                   |
> > | MVNet              | ReCon Transformer | ShapeNet          | **90.5**               |

---

> > > ### Author Response · Authors · 2023-11-21
> > > **Official Comment by Authors (Part 3)**
> > >
> > > **Q: The comparisons in Table 3 are unfair.**
> > >
> > > We thank the reviewer for pointing this out. To address your concern, we have included results pre-trained on ShapeNet in the Official Comment (Part 2). And in Table 3, MVNet-B uses the same Transformer backbone as other methods listed, ensuring a fair comparison. MVNet-L, which utilizes a larger backbone, is presented to demonstrate the scalability of our method – an aspect often overlooked in previous research.
> > >
> > > **Q: There are repeated citations in the bibliography.**
> > >
> > > We thank the reviewer for pointing this out! The issue has been fixed in the updated submission.
> > >
> > > **Q: The result of ScanObjectNN in Figure 1 is different from the result in Table 3.**
> > >
> > > We thank the reviewer for highlighting this observation. The result for ScanObjectNN presented in Figure 1 reflects the classification accuracy of MVNet-L in the ScanObjectNN PB-T50-RS task, which is 91.0. This is indeed consistent with the result reported in Table 3. We will clarify this in our revised manuscript.
> > >
> > > **Q: Why not directly use the transformation matrix.**
> > >
> > > We appreciate the reviewer for raising this intriguing question. In our experiments, we did explore the possibility of predicting the transformation matrix.  However, we found that predicting dense correspondences yielded more accurate results than predicting the transformation matrix directly. This is likely due to the fact that the correspondences are more straightforward for the network to learn from the visual cues than the SE(3) transformation parameters. Furthermore, pre-training with the prediction of the transformation matrix demonstrated inferior results compared to predicting correspondence. We included an ablation study on the ScanNet semantic segmentation task to illustrate this point. The mIoU results, presented in the table below, clearly indicate that predicting dense correspondences during pre-training captures superior features.
> > >
> > > | Multi-view Consistency Module | ScanNet mIoU |
> > > | ----------------------------- | ------------ |
> > > | Transformation matrix         | 75.0         |
> > > | Dense corrspondences          | 75.6         |
> > >
> > > **Q: Is voting strategy employed in the classification of object point clouds?**
> > >
> > > We did not apply any voting strategies on point cloud classification tasks for fair comparison.
> > >
> > > **Q: Why SAM yields inferior performance as a 2D encoder compared to DINOv2.**
> > >
> > > We thank the reviewer for the question! Unfortunately, we are unable to provide a theoretical explanation to the difference in performance. However, we highlight that existing works [3] [4] have also successfully recovered dense correspondences from DINO and DINOv2.
> > >
> > >
> > >
> > > [3] Amir, Shir, Yossi Gandelsman, Shai Bagon, and Tali Dekel. "Deep vit features as dense visual descriptors." arXiv preprint arXiv:2112.05814 2, no. 3 (2021): 4.
> > >
> > > [4] Zhang, Junyi, Charles Herrmann, Junhwa Hur, Luisa Polania Cabrera, Varun Jampani, Deqing Sun, and Ming-Hsuan Yang. "A Tale of Two Features: Stable Diffusion Complements DINO for Zero-Shot Semantic Correspondence." arXiv preprint arXiv:2305.15347 (2023).

---

> > > > ### Comment · Reviewer_GBMb · 2023-11-23
> > > >
> > > > I thank the authors for their great efforts in this work, which address my concerns to some extent. I will keep my positive rating.

---

> > > > > ### Author Response · Authors · 2023-11-23
> > > > > **Thank you for your positive rating!**
> > > > >
> > > > > Thank you for the positive rating! We are delighted to know that our responses have addressed your concerns. We also deeply appreciate the time and effort you invested in evaluating our work. Your feedback and insights are highly valued and greatly appreciated!

---

### Official Review · Reviewer_RdJt · 2023-11-01

**Soundness:** 3 good
**Presentation:** 4 excellent
**Contribution:** 2 fair
**Rating:** 6
**Confidence:** 5

**Summary:**

This paper proposes a new learning framework for point cloud pre-training. The core motivation is to learn a view-consistent 3D feature representation. To enhance the pre-training effectiveness, the authors introduce pre-trained 2D image learning networks for knowledge transfer and also develop an auxiliary task of building pixel-space correspondences. Experiments on various downstream tasks demonstrate the effectiveness of the proposed pre-training method.

**Strengths:**

(1) The paper is well-written and easy to follow. The working mechanism and motivation of each module is clearly explained.

(2) The proposed learning pipeline and pretext task are technically sound.

(3) The resulting supervised fine-tuning performances are encouraging.

**Weaknesses:**

(1) In practice, we may adopt various backbone models for point cloud learning. This requires a generic pre-training scheme that can be used to enhance the feature extraction capability of various point cloud backbones. However, in the manuscript, only the SR-UNet backbone is involved. Hence, the authors should explore more types of backbones for pre-training to validate the proposed method. Note that it is not convincing enough to only give a few results in Table 2 of the supplementary material without detailed explanations and more comprehensive experiments.

(2) Furthermore, the experimental comparison with previous point cloud pre-training approaches is problematic because the adopted backbone models differ. To evaluate the effectiveness of the proposed pre-training framework, the authors should make sure that the performance gains are not from the stronger backbone.

(3) For scene-level experiments, the authors perform pre-training on ScanNet and fine-tune on ScanNet, S3DIS, and SUN RGB-D. In fact, the pre-training dataset and the fine-tuning datasets are all indoor room data, with small domain gaps. Therefore, the actual transferability, which is known to be a key consideration factor for a pre-training method, is questionable. Some cross-domain verifications are needed, such as fine-tuning on some outdoor datasets.

 (4) For object-level experiments, the pre-training dataset Objaverse (with 800K objects) is much larger than ShapeNet/ModelNet used in previous pre-training methods. Therefore, the performance comparison is unfair.

**Questions:**

For different experiments, the authors are also suggested to report their results of training from scratch, such that readers can better observe the relative gains after pre-training the backbone using the proposed method.

---

> ### Author Response · Authors · 2023-11-21
> **Official Comment by Authors (Part 1)**
>
> We express our sincere gratitude to the reviewer for the insightful feedback. In response to the concerns raised, we offer the following clarifications and additional information.
>
> **Q: Adopt various backbone models for point cloud learning.**
>
> We deeply appreciate the reviewer for the insightful suggestion. We totally agree that a generic pre-training scheme should have the ability to enhance feature extraction in various point cloud backbones. In our paper, we have employed SR-UNet as the primary backbone for scene-level experiments and Transformer for shape-level tasks. We contend that this demonstrates the adaptability of our pre-training scheme to at least two different point cloud backbones.
>
> To further validate the efficacy of our method, we have conducted additional experiments incorporating other encoder backbones for the rebuttal. We selected two point cloud encoders for this purpose:
>
> 1. PointNet++-based encoder: For this encoder type, network features are aggregated through multiple set abstraction levels. We utilized PointNeXt [1] as the representative encoder backbone in this category, which is an advanced version of PointNet++.
> 2. Graph-based encoder: In this approach, each point in a point cloud is treated as a graph vertex. Directed edges are generated based on the neighboring points. For this category, we selected the commonly-used DGCNN [2] as our test encoder backbone.
>
> The results, as detailed in the following table, clearly demonstrate significant improvements in both encoder backbones when pre-trained using our method.
>
> | Method             | Backbone  | S3DIS Area 5 mIoU | S3DIS 6-Fold mIoU | ScanNet mIoU |
> | ------------------ | --------- | ----------------- | ----------------- | ------------ |
> | Train from scratch | DGCNN     | 47.9              | 56.1              | 50.2         |
> | MVNet              | DGCNN     | **61.5**          | **66.2**          | **63.0**     |
> | Train from scratch | PointNeXt | 70.8              | 74.9              | 71.5         |
> | MVNet              | PointNeXt | **73.1**          | **78.0**          | **75.2**     |
> | Train from scratch | SR-UNet   | 68.2              | 73.6              | 72.2         |
> | MVNet              | SR-UNet   | **73.8**          | **78.3**          | **75.6**     |
>
> [1] Qian, Guocheng, et al. "Pointnext: Revisiting pointnet++ with improved training and scaling strategies."  In NeuIPS 2022
>
> [2] Wang, Yue, et al. "Dynamic graph cnn for learning on point clouds." *ACM Transactions on Graphics (tog)* 38.5 (2019): 1-12.

---

> > ### Author Response · Authors · 2023-11-21
> > **Official Comment by Authors (Part 2)**
> >
> > **Q: The experimental comparison with previous point cloud pre-training approaches is problematic.**
> >
> > We are grateful to the reviewer for highlighting this aspect. However, we maintain that we have diligently ensured a fair comparison in our study.
> >
> > In the scene-level semantic segmentation task, as shown in Table 1 of the main paper, we selected SR-UNet as the encoder backbone, which is also commonly used in most recent works [3] [4] [5]. For other more earlier related works used DGCNN as the backbone, we show the relevant results in the following table.
> >
> > | Method             | Backbone | S3DIS Area 5 mIoU | S3DIS 6-Fold mIoU | ScanNet mIoU |
> > | ------------------ | -------- | ----------------- | ----------------- | ------------ |
> > | Train from scratch | DGCNN    | 47.9              | 56.1              | 50.2         |
> > | Jigsaw             | DGCNN    | 52.1              | 56.6              | -            |
> > | OcCo               | DGCNN    | 55.4              | 58.5              | -            |
> > | STRL               | DGCNN    | 51.8              | 57.1              | -            |
> > | MVNet              | DGCNN    | **61.5**          | **66.2**          | **63.0**     |
> >
> > For the scene-level detection task, our methodology is consistent with the current standard in the field, as outlined in [3] [4] [6]. We utilize SR-UNet as the encoder, coupled with VoteNet for the detection architecture. This configuration enabled us to demonstrate notable improvements under an equivalent setup. However, as we mentioned in Section 4.2, *"VoteNet’s performance lags behind the state-of-the-art. Therefore, we also evaluate our approach on CAGroup3D, which offers a stronger decoder architecture and delivers state-of-the-art 3D detection performance."* It is important to note that in the CAGroup3D experiment, the only modification was the decoder architecture; the encoder backbone remained SR-UNet. We will provide further clarification on this aspect in our revised manuscript.
> >
> > In the shape-level task, as presented in Table 3, all the methods compared utilize the same Transformer as the encoder backbone. To ensure a fair comparison, we also used the same Transformer as the encoder backbone, referred to as 'MVNet-B'. Under this consistent setting, our method demonstrates significant improvement.
> >
> > In conclusion, we believe that our comparisons across different tasks have been conducted with fairness and in alignment with current standards in the field.
> >
> > [3] Xie, Saining, et al. "Pointcontrast: Unsupervised pre-training for 3d point cloud understanding." In ECCV 2020.
> >
> > [4] Zhang, Zaiwei, et al. "Self-supervised pretraining of 3d features on any point-cloud." In ICCV 2021.
> >
> > [5] Hou, Ji, et al. "Exploring data-efficient 3d scene understanding with contrastive scene contexts." In CVPR 2021.
> >
> > [6] Chen, Yujin, Matthias Nießner, and Angela Dai. "4dcontrast: Contrastive learning with dynamic correspondences for 3d scene understanding." In ECCV 2022.

---

> > > ### Author Response · Authors · 2023-11-21
> > > **Official Comment by Authors (Part 3)**
> > >
> > > **Q: Cross-domain transfer learning.**
> > >
> > > We appreciate the reviewer's suggestion. However, we contend that the domain gap between indoor and outdoor datasets is substantial.
> > >
> > > Firstly, the point clouds from outdoor datasets, typically sourced from LiDAR, are sparse and noisy. Conversely, indoor dataset point clouds, often derived from RGB-D sensors, are much denser.
> > >
> > > Secondly, the types of objects found in indoor and outdoor datasets differ significantly, which compounds the challenges of transfer learning. Objects commonly found in outdoor datasets are usually absent in indoor environments.
> > >
> > > However, in response to the reviewer's query, we conducted additional experiments to assess the cross-domain transferability at shape-level and scene-level.
> > >
> > > The results, detailed in the following table, showcase our methodology. For the S3DIS Area 5 semantic segmentation task, we used SR-UNet as the network architecture. The model was initially pre-trained on the object-level Objaverse dataset and subsequently fine-tuned on the scene-level S3DIS semantic segmentation task. Although not matching the efficacy of scene-level pretraining, object-level pre-training still yielded a noticeable improvement over training from scratch (70.7 mIoU vs 68.2 mIoU).
> > >
> > > Conversely, for the shape-level ScanObjectNN PB-T50-RS classification task, we utilized Transformer as the network architecture. Post pre-training on the ScanNet dataset, this approach also demonstrated a clear improvement compared to training from scratch.
> > >
> > > | Task                                  | Pre-train dataset | Acc/mIoU |
> > > | ------------------------------------- | ----------------- | -------- |
> > > | S3DIS Area 5 Sem Segmentation         | None              | 68.2     |
> > > |                                       | Objaverse         | 70.7     |
> > > |                                       | ScanNet           | 73.8     |
> > > | ScanObjectNN PB-T50-RS Classification | None              | 77.2     |
> > > |                                       | Objaverse         | 87.8     |
> > > |                                       | ScanNet           | 84.9     |
> > >
> > > Table: Cross-Domain Transfer Learning between shape-level and scene-level tasks. For the semantic segmentation task, we report mIoU on the S3DIS Area 5. For ScanObjectNN PB-T50-RS classification task, we report classification accuracy.
> > >
> > >
> > >
> > > **Q: Objaverse (with 800K objects) is much larger than ShapeNet.**
> > >
> > > We are grateful to the reviewer for highlighting this aspect. In response, we pre-trained our model on the ShapeNet dataset, and show the ScanObjectNN PB-T50-RS classification results in the table below. Our model demonstrates superior performance in this context.
> > >
> > > | Method             | Backbone    | Pre-train dataset | ScanObjectNN PB-T50-RS |
> > > | ------------------ | ----------- | ----------------- | ---------------------- |
> > > | Train from scratch | Transformer | -                 | 77.2                   |
> > > | PointBERT          | Transformer | ShapeNet          | 83.1                   |
> > > | Point-MAE          | Transformer | ShapeNet          | 85.2                   |
> > > | ULIP               | Transformer | ShapeNet          | 86.4                   |
> > > | MVNet-B            | Transformer | ShapeNet          | **86.7**               |
> > >
> > > **Q: Report training from scratch results.**
> > >
> > > We thank the reviewer for this insightful suggestion! We have included the results of training from scratch in Table 3 and will incorporate the training from scratch results in Tables 1 and 2 in our revised manuscript.

---

> > > > ### Comment · Reviewer_RdJt · 2023-11-22
> > > >
> > > > Thanks for the authors' detailed explanations and extensive experiments. My raised concerns have been basically addressed in the response. I will correspondingly update my rating.

---

> > > > > ### Author Response · Authors · 2023-11-22
> > > > > **Thank you for your response!**
> > > > >
> > > > > Thank you for your feedback and for improving the score! We are glad that our responses have addressed your concerns! Thank you again for the time and effort you have dedicated to reviewing our work!

---

### Author Response · Authors · 2023-11-21

We appreciate all the reviewers for their hard work! Please find our responses to your individual questions below. We look forward to discussing any issues further should you have any follow-up concerns!

---

### Meta-Review · Area_Chair_fQxe · 2023-12-05

**Metareview:**

The paper proposes a 3D point cloud pre-training method with knowledge distillation from pre-trained 2D models and a multi-view consistency loss. Evaluations on both scene-level and object-level 3D tasks demonstrate the effectiveness of the proposed approach.
Initially, the reviewers had concerns about the generalization to different point cloud backbones, unfair comparison with prior works, missing comparisons with more recent works and no zero-shot experiments in the object-level.
The rebuttal provided more experimental results and addressed most of the reviewers’ concerns. The reviewers unanimously recommend acceptance, and the AC agrees.
The authors should add the new results and clarifications in the rebuttal to the final version and also are encouraged to add zero-shot experiments.

**Justification For Why Not Higher Score:**

The paper shares similar ideas with prior works such as ULIP and ReCon to distill knowledge from pre-trained 2D models for 3D point cloud pre-training. The technical contribution is not big enough as mentioned by the reviewers, and the performance improvements on the object-level evaluation are marginal.

**Justification For Why Not Lower Score:**

The simplicity is a merit of the proposed method. The evaluations on both scene-level and object-level tasks demonstrate the effectiveness of the approach.

---

### Decision · Program_Chairs · 2024-01-16

Accept (poster)